# The Nonlinearity Coefficient - Predicting Generalization in Deep Neural Networks

## Abstract

For a long time, designing neural architectures that exhibit high performance was considered a dark art that required expert hand-tuning. One of the few well-known guidelines for architecture design is the avoidance of exploding or vanishing gradients. However, even this guideline has remained relatively vague and circumstantial, because there exists no well-defined, gradient-based metric that can be computed *before* training begins and can robustly predict the performance of the network *after* training is complete.

We introduce what is, to the best of our knowledge, the first such metric: the nonlinearity coefficient (NLC). Via an extensive empirical study, we show that the NLC, computed in the network's randomly initialized state, is a powerful predictor of test error and that attaining a right-sized NLC is essential for attaining an optimal test error, at least in fully-connected feedforward networks. The NLC is also conceptually simple, cheap to compute, and is robust to a range of confounders and architectural design choices that comparable metrics are not necessarily robust to. Hence, we argue the NLC is an important tool for architecture search and design, as it can robustly predict poor training outcomes before training even begins.

## 1 Introduction

Designing neural architectures that perform well can be a difficult process. In particular, the exploding / vanishing gradient problem has been a major challenge for building very deep neural networks at least since the advent of gradient-based parameter learning (Hochreiter, 1991; Hochreiter & Schmidhuber, 1997; Bengio et al., 1994). However, there is still no consensus about which metric should be used for determining the presence of pathological exploding or vanishing gradients. Should we care about the length of the gradient vector (He et al., 2015), or about the size of individual components of the gradient vector (Schoenholz et al., 2017; Yang & Schoenholz, 2017; Glorot & Bengio, 2010), or about the eigenvalues of the Jacobian (Saxe et al., 2014; Pascanu et al., 2013; Pennington et al., 2017)? Depending on the metric used, different strategies arise for combating exploding and vanishing gradients. For example, manipulating the width of layers as suggested by e.g. Yang & Schoenholz (2018); Han et al. (2017) can greatly impact the size of gradient vector components but tends to leave the length of the entire gradient vector relatively unchanged. The popular He initialization for ReLU networks (He et al., 2015) is designed to stabilize gradient vector length, wheareas the popular Xavier initialization for tanh networks (Glorot & Bengio, 2010) is designed to stabilize the size of gradient vector components. While the papers cited above provide much evidence that gradient explosion / vanishing when defined according to some metrics is associated with poor performance when certain architectures are paired with certain optimization algorithms, it is often unclear how general those results are.

We make the following core contributions.

1. We introduce the *nonlinearity coefficient (NLC)*, a gradient-based measurement of the degree of nonlinearity of a neural network (section 3).

2. We show that the NLC, computed in the networks randomly initialized state, is a powerful predictor of test error and that attaining a right-sized NLC is essential for achieving an optimal test error, at least in fully-connected feedforward networks (section 4).

3. We explain that, by design, the NLC is not susceptible to a range of confounders that render many other metrics unreliable, such as changes to input scale, width and bias (section 6).

We demonstrate the properties of the NLC via an extensive empirical study covering a wide range of network architectures (section 4). The scope of our experiments exceeds that of the vast majority of related work. We conduct 43.000 full training runs. As the NLC is also conceptually simple and cheap to compute, it is a useful guide for architecture design and search. Architectures with a sub-optimal NLC can be discarded a priori and computational resources don't have to be spent on training them.

The NLC (defined at the top of page 3) combines the Frobenius norm of the Jacobian of a neural network with the global variability of the input data and the global variability of the network's outputs into a single metric. Despite its simplicity, it is tied to many important properties of the network. It is a remarkably accurate predictor of the network's nonlinearity as measured by the relative diameter of the regions in input space that can be well-approximated by a linear function (section 3 and figure 1). It is closely related to the nonlinearity of the individual activation functions used in the network and the degree to which they can be approximated by a linear function (section 5). It is tied to the susceptibility of the network's output to small random input perturbations.

## 2 NOTATION AND TERMINOLOGY

We define a neural network $f$ as a function of the input $x$. Both $x$ and the output $f(x)$ are vectors of fixed dimensionality $d_{\text{in}}$ and $d_{\text{out}}$ respectively. We assume a prediction framework, where the output is considered to be the prediction and the goal is to minimize the value of the 'error' $e$ over this prediction and the label $y$, in expectation over some data distribution $\mathcal{D}$. I.e., we wish to minimize $\mathbb{E}_{(x,y)\sim\mathcal{D}}[e(f(x),y)]$. In this paper, $e$ is always classification accuracy. During training, we replace $\mathcal{D}$ with the training set and $e$ with the surrogate loss function $\ell$, which in this paper is always softmax plus cross-entropy. Let $\mathcal{J}(x) := \frac{df(x)}{dx}$ be the Jacobian of the output with respect to the input. $x(i)$ and $f(x,j)$ denote the $i$'th component of the input vector and the $j$'th component of the output vector respectively, where $1 \le i \le d_{in}$ and $1 \le j \le d_{out}$. Denote the component-wise standard deviation of a (possibly multivariate) random variable $X$ as $\mathbb{S}[X]$. Finally, let the 'quadratic expectation' $\mathbb{Q}$ of a random variable $X$ be defined as $\mathbb{Q}[X] = \sqrt{(\mathbb{E}[X^2])}$, i.e. the generalization of the quadratic mean to random variables.

## 3 ON THE NONLINEARITY OF NEURAL NETWORKS - DERIVING THE NLC

Consider an arbitrary neural network $f$ and assume it has a well-defined bounded domain $\mathbb{D}$ and bounded co-domain $\mathbb{F}$. Then the Jacobian $\mathcal{J}$ taken at some $x \in \mathbb{D}$ defines a local linear approximation of the function $f$ around $x$, i.e. $f(x+\delta) \approx f(x) + \mathcal{J}(x)\delta$ for sufficiently small $\delta$. The key insight behind the NLC and ultimately behind this paper is that there is a simple criterion for determining whether the approximation can be accurate for a given $\delta$: If $f(x) + \mathcal{J}(x)\delta$ is far away from $\mathbb{F}$, then because $f(x+\delta) \in \mathbb{F}$, $f(x) + \mathcal{J}(x)\delta$ is far away from $f(x+\delta)$ and thus it is inaccurate. We can use this insight to establish an approximate bound for the size of $\delta$ by referencing a well-known property of the Frobenius norm.

**Proposition 1.** *Let $A$ be an $m \times n$ matrix and $u$ a random $n$-dimensional vector of fixed length and uniformly random orientation. Then $\frac{||A||_F}{\sqrt{n}}||u||_2 = \mathbb{Q}_u||Au||_2$. (See section F for the short proof.)*

As we also have $f(x) \in \mathbb{F}$, we deduce that we must have $\frac{||\mathcal{J}(x)||_F}{\sqrt{d_{in}}}||\delta||_2 \lesssim \text{diameter}(\mathbb{F})$ for the local linear approximation at $x$ to be accurate for a randomly oriented $\delta$. In fact, an approximate bound for the diameter of the linearly approximable region at $x$ in a random direction is $\frac{\sqrt{d_{in}}\text{diameter}(\mathbb{F})}{||\mathcal{J}(x)||_F}$. However, the size of this diameter by itself is not sufficient to judge the nonlinearity of $f$, because it does not take into account the size of the domain $\mathbb{D}$. If $\mathbb{D}$ fits entirely within the linearly approximable region, then $f$ is close to a linear function. If $\mathbb{D}$ is large compared to the linearly approximable region, we consider $f$ highly nonlinear at $x$. Hence, we consider instead the approximate bound for the *relative diameter of the linearly approximable region* in a random direction, $\frac{\sqrt{d_{in}}\text{diameter}(\mathbb{F})}{||\mathcal{J}(x)||_F\text{diameter}(\mathbb{D})}$. We posit the inverse of this value, $\frac{||\mathcal{J}(x)||_F\text{diameter}(\mathbb{D})}{\sqrt{d_{in}}\text{diameter}(\mathbb{F})}$, as our nonlinearity metric for $f$ at $x$.

Of course, in practice, a network might not have a well-defined, bounded domain and co-domain. Here we make a final modeling assumption. We proxy the diameter with the quadratic expectation of the distance of two random points from the data distribution. Thus our nonlinearity estimate becomes $\frac{||\mathcal{J}(x)||_F \mathbb{Q}_{x,x'\sim\mathcal{D}}||x-x'||_2}{\sqrt{d_{in}}\mathbb{Q}_{x,x'\sim\mathcal{D}}||f(x)-f(x')||_2}$. And as we further show in section F, the quadratic expectation of this equals the NLC as defined below.

**Definition 1.** The *nonlinearity coefficient (NLC)* of a network $f$ and inputs $x \sim \mathcal{D}$ is
$$NLC(f,\mathcal{D}) := \frac{\left(\mathbb{Q}_x||\mathcal{J}(x)||_F\right)\left(\mathbb{Q}_i(\mathbb{S}_x x(i))\right)}{\sqrt{d_{\text{out}}}\mathbb{Q}_j(\mathbb{S}_x f(x,j))}.$$

The terms $\mathbb{S}_x x(i)$ and $\mathbb{S}_x f(x,j)$ denote the standard deviation of the activation of the $i$'th input neuron and $j$'th output neuron under the data distribution respectively. $\mathbb{Q}_i(\mathbb{S}_x x(i))$ and $\mathbb{Q}_j(\mathbb{S}_x f(x,j))$ denote the quadratic expectation of these values across all input / output neurons respectively. We also have $\mathbb{Q}_i(\mathbb{S}_x x(i)) = \sqrt{\frac{\text{Tr}(C_x)}{d_{\text{in}}}}$ and $\mathbb{Q}_j(\mathbb{S}_x f(x,j)) = \sqrt{\frac{\text{Tr}(C_f)}{d_{\text{out}}}}$, where $C_x$ and $C_f$ are the covariance matrices of $x$ and $f(x)$ under the data distribution. Hence, the NLC can be re-written as $\sqrt{\frac{\text{Tr}(\mathbb{E}_x \mathcal{J}\mathcal{J}^T)\text{Tr}(C_x)}{d_{\text{in}}\text{Tr}(C_f)}}$. As a sanity check, let's look at the NLC of a linear network. In that case, $f(x) = Ax + b$ for some $A$ and $b$. Then the NLC becomes $\sqrt{\frac{\text{Tr}(AA^T)\text{Tr}(C_x)}{d_{\text{in}}\text{Tr}(AC_xA^T)}}$. This value equals 1, for example, if $C_x$ is a multiple of the identity or $A$ is orthogonal. We further conjecture that this value is close to 1 for large, random matrices $A$ as they occur in practice in randomly initialized neural networks, though this analysis goes beyond the scope of this paper.

Note that an alternative interpretation of the NLC is that it represents the expected sensitivity of the network output with respect to small, randomly oriented changes to the input, normalized by the global variability of the input, output and $\sqrt{d_{\text{out}}}$. Finally, we refer readers interested in a pictorial illustration of the NLC to section A in the appendix.

**Computing the NLC** It is worth noting that the NLC is cheap to (approximately) compute. Throughout this study, we compute $\mathbb{Q}_i(\mathbb{S}_x x(i))$ and $\mathbb{Q}_j(\mathbb{S}_x f(x,j))$ exactly via a single pass over the training set. $\mathbb{Q}_x||\mathcal{J}(x)||_F$ can be computed stochastically by backpropagating Gaussian random vectors from the output layer. See section G for details.

## 4 ON THE PREDICTIVE POWER OF THE NLC - LARGE-SCALE STUDY

Now we investigate the empirical properties of the NLC through a large-scale study.

**Architectures used** We sampled architectures at random by varying the depth of the network, the scale of initial weights, scale of initial bias, activation function, normalization method, presence of skip connections, location of skip connections and strength of skip connections. We chose from a set of 8 activation functions (table 1), which were further modified by random dilation, lateral shift and debiasing. For now, we only considered fully-connected feedforward networks, as is (perhaps unfortunately) common in analytical studies of deep networks (e.g. Saxe et al. (2014); Balduzzi et al. (2017); Schoenholz et al. (2017)). We have no reasons to suspect our results will not generalize to CNNs, and we plan to investigate this point in future work. See section C for the full details of our architecture sampling scheme.

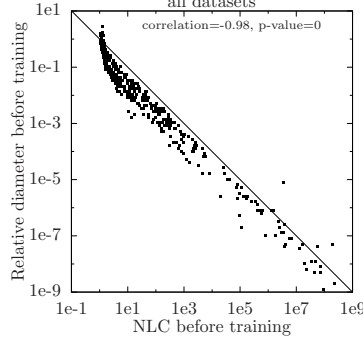

Figure 1: NLC vs the relative diameter of linearly approximable regions. See section E.1 for details.

**Datasets used** We studied three datasets: MNIST, CIFAR10 and waveform-noise. All our results were highly consistent across these datasets. waveform-noise is from the UCI repository of datasets popular for evaluating fully-connected networks (Klambauer et al., 2017). See section D for further details on dataset selection and preprocessing. We sampled 250 architectures per dataset, a total of 750.

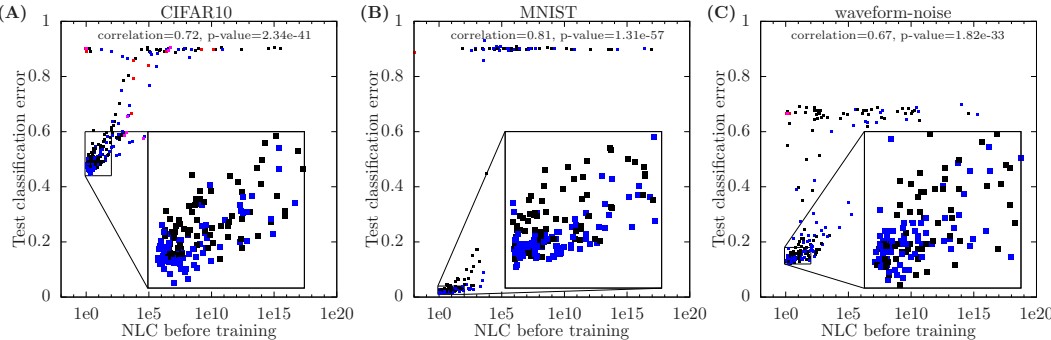

Figure 2: NLC versus test error. Points shown in red correspond to architectures with high output bias ($1000\mathbb{Q}_j(\mathbb{S}_x f(x,j)) < \mathbb{Q}_{j,x} f(x,j)$). Points shown in blue correspond to architectures that have skip connections. Inset graphs in the bottom right are magnifications of the region $0.8 < NLC < 100$. See section E.2 for details.

**Training protocol**    We trained each architecture with SGD with 40 different starting learning rates and selected the optimal one via a held-out validation set, independently for each architecture. During each run, we reduced the learning rate 10 times by a factor of 3. All training runs were conducted in 64 bit floating point. See section E for further experimental details and section B.1 for an analysis of how the learning rate search and numerical precision contributed to the outcome of our study.

**Presentation of results**    The results of this study are shown in figures 1, 2, 3, 4, 7, 8, 9 and 10. All figures except figure 7 are scatter plots where each point corresponds to a single neural architecture. In most graphs, we show the correlation and its statistical significance between the quantities on the x and y axis at the top. Note that if any quantity is plotted in log-scale, the correlation is also using the log of that quantity. For each architecture, we only studied a single random initialization. Given a limited computational budget, we preferred studying a larger number of architectures instead. Note that all values depicted that were computed after training, such as test error or 'NLC after training' in figure 3C, are based on the training run with the lowest validation error, as described above.

**The NLC measures nonlinearity**    First, we verify that the NLC is indeed an accurate measure of nonlinearity. In figure 1, we plot the relative diameter of the linearly approximable regions of the network as discussed in section 3 and defined in section E.1 against the NLC in the randomly initialized state. We find a remarkably close match between both quantities. This shows empirically that our informal derivation of the NLC in section 3 leads to remarkably accurate nonlinearity estimates.

**The NLC predicts test error**    In figure 2, we plot the NLC computed in the randomly initialized state, *before* training, against the test error *after* training. We find that for all three datasets, the test error is highly related to the NLC. Further, figure 2 indicates that one must start with an NLC in a narrow range, approximately between 1 and 3, to achieve optimal test error, and the further one deviates from that range, the worse the achievable test error becomes. It is worth noting that some architectures, despite having an NLC in or close to this range, performed badly. One cause of this, high output bias, is explored later in this section. To verify that our results were not dependent on using the SGD optimizer, we re-trained all 250 waveform-noise architectures with Adam using the same training protocol. In figure 3F, we find that the results closely match those of SGD from figure 2C. A caveat is that we do not currently have a way to detect the ideal NLC range for a given dataset, except through observation, though we find this range to be consistent across our three datasets.

**NLC after training**    In figure 3B, we plot the value of the NLC before training versus after training. Both values were computed on the training set. We find that for the vast majority of architectures, the value of the NLC decreases. However, if the NLC is very large in the beginning, it remains so. Overall, the before-training NLC significantly predicts the after-training NLC. In figure 3C, we plot the after-training NLC versus test error. We find that unless the NLC lies in a narrow range, test error is close to random. Interestingly, the after-training NLC has a significantly lower correlation with test error than the before-training NLC.

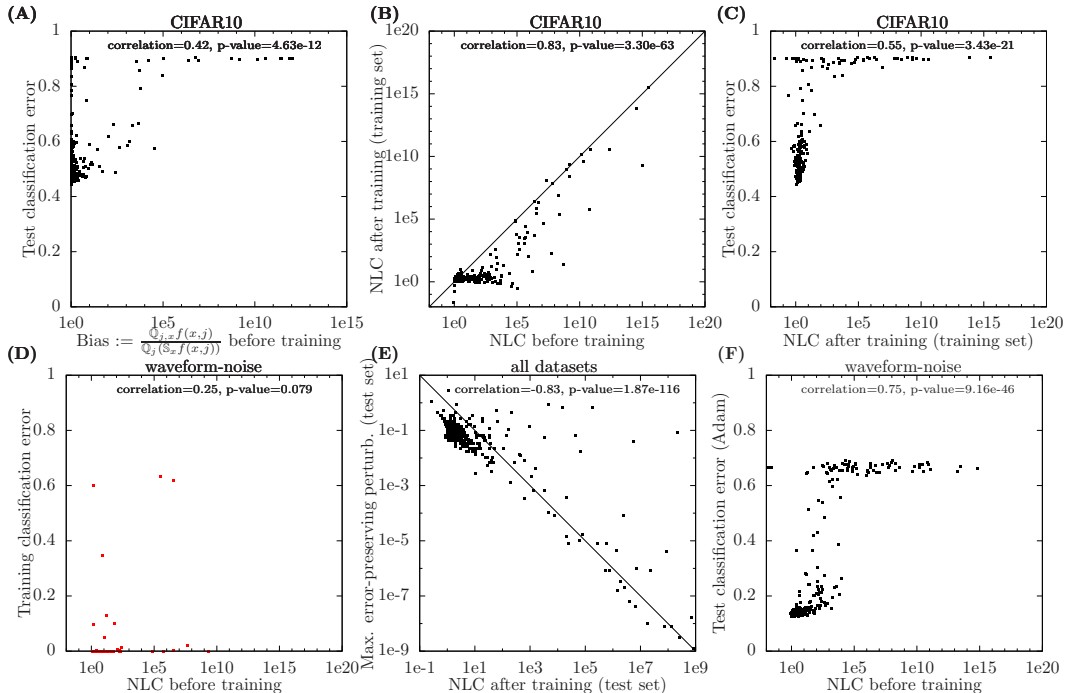

Figure 3: Detailed results from our empirical study. See main text for explanation and section E.2 (figures A/B/C/D/F) and section E.3 (figure E) for further details.

**NLC predicts generalization, not trainability** In figure 3D, we show the training error achieved for 50 randomly selected architectures for waveform-noise. We re-trained those architectures without using early stopping based on the validation error and considered an even larger range of starting learning rates. We depict the lowest training classification error that was achieved across all learning rates. Points are shown in red for visibility. We find that independently of the NLC, the vast majority of architectures achieve a zero or near-zero training error. This finding is the opposite of that of Schoenholz et al. (2017) and Xiao et al. (2018), who claim that networks which are highly sensitive to small input perturbations are untrainable. The reason we were able to train these architectures was our extensive learning rate search as well as our decision to train with 64 bit precision. In fact, we found that architectures with high NLC often require very small learning rates and very small parameter updates to train successfully. One architecture required a learning rate as small as 5e-18! See section B.1 for further analysis on this point. In summary, we find that many architectures for which the NLC correctly predicts high test error are nonetheless trainable. Of course, one might expect that a high sensitivity leads to poor generalization. As a sanity check, we conducted an experiment where we corrupted the test set with small random perturbations and measured how large these perturbations could be before the test error increased significantly. We plot this in figure 3E. As expected, for the majority of high-NLC architectures, labels can be corrupted and the error increased with incredibly small perturbations.

**Summary** We interpret the results observed so far as follows. To generalize, the network must attain a critical NLC after training. This is only possible in practice if the initial NLC is already close. In that case, the networks often learns automatically to adopt a more ideal NLC. However, unless the initial NLC is itself in the critical range, we cannot attain optimal performance.

**Further predictors: bias and skip connections** In figure 2, we mark in red all points corresponding to architectures that have a very biased output, i.e. $1000\mathbb{Q}_j(\mathbb{S}_x f(x,j)) < \mathbb{Q}_{j,x} f(x,j)$. We note that all of these architecture attain a high test error, even if their NLC is small. In figure 3A, we plot the output bias before training against test error. We find that indeed, to achieve an optimal test error, a low initial bias is required. In section B.2, we further show that just as the NLC, the bias also tends to decline throughout training and that attaining a very low bias after training is even more essential.

| | ReLU | SELU | tanh | sigmoid | even tanh | Gaussian | square | odd square |
|---|---|---|---|---|---|---|---|---|
| Formula | $\max(s,0)$ | 1[1] | $\tanh(s)$ | $\frac{1}{1+e^{-s}}$ | $\tanh(\lvert s\rvert)$ | $\frac{1}{\sqrt{2\pi}}e^{-\frac{s^2}{2}}$ | $s^2$ | $s*\lvert s\rvert$ |
| Illustration | | | | | | | | |
| (A) $NLC_\tau(1)$ | 1.211 | 1.035 | 1.085 | 1.017 | 2.335 | 1.577 | 1.414 | 1.155 |
| (B) NLC depth 2 | 1.22 | 1.05 | 1.09 | 1.03 | 2.34 | 1.61 | 1.48 | 1.20 |
| (C) $NLC_\tau(1)^{48}$ | 9793 | 5.21 | 50.19 | 2.25 | $4.76e17$ | $3.13e9$ | $1.67e7$ | 1009 |
| (D) NLC depth 49 | 7376 | 7.06 | 59.9 | 3.06 | $4.90e17$ | $3.74e9$ | $4.28e12$ | $1.38e11$ |
| (E) Lin. app. err. | 0.222 | 0.030 | 0.075 | 0.0024 | 0.276 | 0.155 | 2.000 | 0.178 |

Table 1: Activation functions used in this study with important metrics. See main text for explanation and section E.4 for details.

In figure 2, we plot in blue all points corresponding to architectures that have skip connections. Philipp et al. (2018) argued that skip connections reduce the gradient growth of general architectures as well as make a further contribution to performance. Correspondingly, we find that skip connections lead to lower NLC values and to lower test errors for a given NLC level. To enable a more convenient visual comparison, we plot results for architectures with and without skip connections separately in section B.3.

## 5 ON THE LINEAR APPROXIMABILITY OF ACTIVATION FUNCTIONS

In this section, we expose the connection between the NLC and low-level properties of the activation functions the network uses. Given a 1d activation function $\tau$, we define

$$NLC_\tau(\sigma) := \frac{\sigma}{\mathbb{S}_{s\sim\mathcal{N}(0,\sigma)}\tau(s)}\mathbb{Q}_{s\sim\mathcal{N}(0,\sigma)}\tau'(s)$$

It is easy to check that if the input $x$ is distributed according to a Gaussian with zero mean and covariance matrix $\sigma I$, and $f$ simply applies $\tau$ to each input component, then we have $NLC(f,\mathcal{D}) = NLC_\tau(\sigma)$. Consider a randomly initialized network where each layer is made up of a fully-connected linear operation, batch normalization, and an activation function $\tau$ that is applied component-wise. It turns out that if the network is sufficiently wide, the pre-activations of $\tau$ are approximately unit Gaussian distributed. This follows from the central limit theorem (Poole et al., 2016). Hence, we expect each layer to contribute approximately $NLC_\tau(1)$ to the nonlinearity of the entire network. To verify this, we train a 2-layer network with batchnorm, which contains a single copy of $\tau$ at the single hidden layer. In table 1, we show $NLC_\tau(1)$ for all 8 activation functions we used (line A), as well as the median empirical NLC over 10 random initializations of the 2-layer network (line B). We indeed find a close match between the two values. We then measure the NLC of 49-layer batchnorm networks, which contain 48 activation functions. For 6 out of 8 activation functions, this NLC (line D) closely matches the exponentiated $NLC_\tau(1)^{48}$ (line C). Hence, we find that nonlinearity compounds exponentially and that the NLC of a network is closely tied to which activation function is used. Note that the reason that the NLC value of the 'square' and 'odd square' activation functions diverge from $NLC_\tau(1)^{\text{depth}-1}$ at high depth is because those activation functions are unstable, which causes some inputs to growth with depth whereas the vast majority of inputs collapse to zero.

We then verified that $NLC_\tau$ is a meaningful measure of nonlinearity for an activation function. We computed the best linear fit for each $\tau$ given unit Gaussian input and then measured the ratio of the power of the signal filtered by this best linear fit over the power of the preserved signal. In table 1(line E), we find that for ReLU, SELU, tanh, sigmoid and Gaussian activation functions, there is a close correspondence in that this linear approximation error is around $NLC_\tau(1) - 1$. While this relationship breaks down for the 3 most nonlinear activation functions, their linear approximation error still exceeds those of the other 5. We conclude that $NLC_\tau$ is a meaningful measure of nonlinearity and that the NLC of an architecture can be calibrated by changing the linear approximability of the activation functions used.

---

[1] $\mathbb{1}_{s>0}1.0507s + \mathbb{1}_{s<0}1.75814(e^s - 1)$ (Klambauer et al., 2017)

## 6   ON THE ROBUSTNESS OF THE NLC VS OTHER METRICS - RELATED WORK

In this section, we discuss how the NLC compares against other metrics that are used for predicting test error in the deep learning field. We focus our analysis on five popular and representative examples: the gradient vector components, the size of gradient vector length, the Lipschitz constant, correlation information preservation, and depth. We find that each metric is susceptible to very simple confounders that render them unreliable in practice. The NLC, by design, is invariant to all these confounders.

**Gradient vector component size**   Historically, there has not been a well-accepted metric for determining the presence of pathological exploding or vanishing gradients. Recently, Schoenholz et al. (2017); Yang & Schoenholz (2017); Glorot & Bengio (2010) used the magnitude of gradient vector components for this job. We paraphrase this metric as $\mathbb{Q}_{i,x}\frac{d\ell}{dx}(i)$ and abbreviate it as GVCS. This metric has at least two drawbacks that render it unreliable in practice.

The first drawback is that this metric is confounded by simple multiplicative rescaling. For example, assume we are using a plain network that begins with a linear operation followed by batch normalization or layer normalization (Ba et al., 2016). Then we can re-scale the input data with an arbitrary constant $c$ and not only preserve the output of the network in the initialized state, but the entire trajectory of the parameter during training and therefore the final test error. Yet, multiplying the input by $c$ causes the GVCS to shrink by $c$. Thus we can arbitrarily control GVCS while preserving test error, and therefore GVCS cannot be used as a direct predictor of test error. We observe a similar effect when the network output is re-scaled with a constant $c$. This causes GVCS to grow by $c$. As long as the loss function can handle large incoming values, which softmax+cross-entropy can do at least in some circumstances (Yang & Schoenholz, 2017), we can again control GVCS arbitrarily without compromising performance.

This drawback shows up in even more insidious ways. Consider a plain network with activation functions of form $\tau(s) = 1 + \frac{1}{k}\sin(ks)$. As $k \rightarrow \infty$, $\tau$ eliminates all meaningful structure in the input data. The NLC converges to infinity to reflect this. Yet, GVCS is stable. Examining the NLC, we find that the increase in nonlinearity is captured by the $\mathbb{Q}_i(\mathbb{S}_x f(x, i))$ term, but not by the gradient. The crux is that the variability of the network output is down-scaled in proportion to the increase in nonlinearity, and this confounds the GVCS. The same effect occurs with He-initialized plain ReLU networks as depth increases.

The second drawback is that the GVCS is also confounded by changing the width of the network. For example, consider a network that begins with a linear operation and has input dimension $d_{in}$. Then we can increase the input dimension by an integer factor $c$ by duplicating each input dimension $c$ times. We can approximately maintain the learning dynamics by reducing the scale of initial weights of the first linear operator by $\sqrt{c}$ and the learning rate for that operator by $c$. Again, this transformation leaves the NLC unchanged but reduces GVCS by $\sqrt{c}$, allowing us again to control GVCS without compromising performance.

**Gradient vector length / Lipschitz constant**   While less popular than GVCS, these two metrics are also used as a predictor for network performance (e.g. He et al. (2015) / Cisse et al. (2017) respectively). Both metrics are susceptible to multiplicative scaling just as GVCS, and the same examples apply. However, in contrast to GVCS, they are not susceptible to a change in input width.

**Correlation information**   Correlation information was recently proposed by Schoenholz et al. (2017); Yang & Schoenholz (2017). They claim that preserving the correlation of two inputs as they pass through the network is essential for trainability, and hence also for a low test error. However, this metric also has an important confounder: additive bias.

Assume we are using a network that employs batchnorm. Then biases in the features of the input do not significantly affect learning dynamics, as this bias will be removed by the first batchnorm operation. Yet, adding a constant $c$ to the input can arbitrarily increase correlation between inputs without affecting the correlation of the outputs. So, again, the degree of correlation change through the network can be increased arbitrarily without altering network performance.

**Depth**   A large body of work has detailed the benefits of depth in neural networks (e.g. Montafur et al. (2014); Delalleau & Bengio (2011); Martens et al. (2013); Bianchini & Scarselli (2014); Shamir & Eldan (2015); Telgarsky (2015); Mhaskar & Shamir (2016)). Most of these works focus on finding specific functions which can be represented easily by deep networks, but require a prohibitively large number of neurons to represent for a shallow network. In figure 4, we plot the test error achieved by our architectures on CIFAR10 against depth. We find that there is actually a positive correlation between both quantities. We suspect this is mainly because deeper networks tend to have a larger NLC. Of course, depth cannot be used as a direct predictor of performance as it does not account for all the confounders discussed throughout this section.

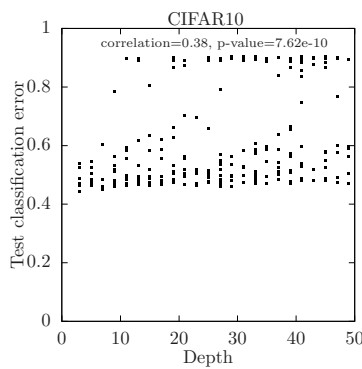

Figure 4: Depth versus test error.

## 7    DISCUSSION AND CONCLUSION

We introduced the nonlinearity coefficient, a measure of neural network nonlinearity that is closely tied to the relative diameter of linearly approximable regions in the input space of the network, to the sensitivity of the network output with respect to small input changes, as well as to the linear approximability of activation functions used in the network. Because of this conceptual grounding, because its value in the randomly initialized state is highly predictive of test error while also remaining somewhat stable throughout training, because it is robust to simple network changes that confound other metrics such as raw gradient size or correlation information, because it is cheap to compute and conceptually simple, we argue that the NLC is the best standalone metric for predicting test error in fully-connected feedforward networks. It has clear applications to neural architecture search and design as it allows sub-optimal architectures to be discarded before training. In addition to a right-sized NLC, we also found that avoiding excessive output bias and using skip connections play important independent roles in performance.

This paper makes important contributions to several long-standing debates. We clearly show that neural networks are capable of overfitting when the output is too sensitive to small changes in the input. In fact, our random architecture sampling scheme shows that such architectures are not unlikely to arise. However, overfitting seems to be tied not to depth or the number of parameters, but rather to nonlinearity. In contrast to Schoenholz et al. (2017); Xiao et al. (2018), we find that a very high output sensitivity does not harm trainability, but only generalization. This difference is likely caused by our very extensive learning rate search and 64 bit precision training.

While the popular guidance for architecture designers is to avoid exploding and vanishing gradients, we argue that achieving an ideal nonlinearity level is the more important criterion. While the raw gradient is susceptible to confounders and cannot be directly linked to meaningful network properties, the NLC captures what appears to be a deep and robust property. It turns out that architectures that were specifically designed to attain a stable gradient, such as He-initialized ReLU networks, in fact display a divergent NLC at great depth.

It has been argued that the strength of deep networks lies in their exponential expressivity (e.g. Raghu et al. (2017); Telgarsky (2015)). While we show that the NLC indeed exhibits exponential behavior, we find this property to be largely *harmful*, not helpful, as did e.g. Schoenholz et al. (2017). While very large datasets likely benefit from greater expressivity, in our study such expressivity only leads to lack of generalization rather than improved trainability. In fact, at least in fully-connected feedforward networks, we conjecture that great depth does not confer significant practical benefit.

In future work, we plan to study whether the ideal range of NLC values we discovered for our three datasets ($1 \lessapprox NLC \lessapprox 3$) holds also for larger datasets and if not, how we might predict this ideal range a priori. We plan to investigate additional causes for why certain architectures perform badly despite a right-sized NLC, as well as extend our study to convolutional and densely-connected networks. We are interested in studying the connection of the NLC to e.g. adversarial robustness, quantizability, sample complexity, training time and training noise. Finally, unfortunately, we found the empirical measurement of the NLC to be too noisy to conclusively detect an underfitting regime. We plan to study this regime in future.

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

| depth | 2 | 5 | 10 | 15 | 20 | 25 | 50 |
|---|---|---|---|---|---|---|---|
| NLC | 1.22 | 2.25 | 5.97 | 15.2 | 37.7 | 95.8 | 9952 |
| Illustration | | | | | | | |

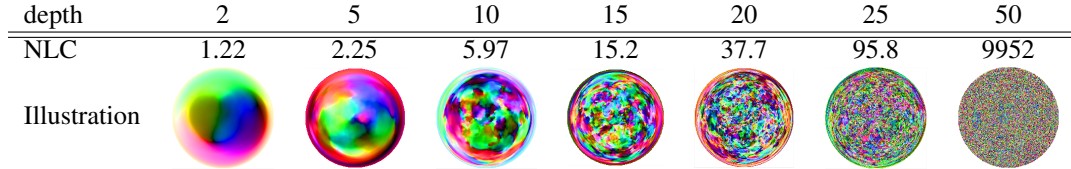

Table 2: Illustration of the function computed by fully-connected batchnorm-ReLU networks at different depths in the randomly initialized state. Each disc represents a 2D subspace of the input space and each color corresponds to a different region of the output space. CIFAR10 was used to compute the NLC.

## A  THE NLC EXPLAINED PICTORIALLY

The goal of this section is to provide an intuitive, graphical explanation of the NLC in addition to the mathematical derivation and analysis in section 3 for readers interested in developing a better intuition of this concept.

### A.1  1D TOY EXAMPLE

In figure 5, we illustrate the meaning of the NLC in the case of an example function $f$ with a single input and output dimension, and a bounded domain $\mathbb{D}$ and co-domain $\mathbb{F}$. $f$ is a simple sin curve, shown in blue. $x_1$ and $x_2$ are two sample inputs. We plot the location of $(x_1, f(x_1))$ in red and the location of $(x_2, f(x_2))$ in olive. The thick red and olive lines correspond to the local linear approximation of $f$ at $x_1$ and $x_2$ respectively, which is simply the tangent line of the blue curve. The shaded olive and red regions correspond to the intervals in which the local linear approximations fall inside the co-domain $\mathbb{F}$.

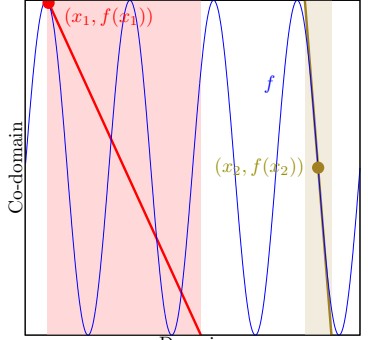

Figure 5: 1d pictorial illustration of the NLC

It is easy to check that the proportion of the domain covered by the red interval and olive interval is $\frac{\text{diameter}(\mathbb{F})}{f'(x_1)\text{diameter}(\mathbb{D})}$ and $\frac{\text{diameter}(\mathbb{F})}{f'(x_2)\text{diameter}(\mathbb{D})}$ respectively. The insight behind the NLC is that both linear approximations can only be accurate while they remain inside their respective shaded area, or at least close to it. This is evidently true in both cases, as both tangent lines quickly move away from the co-domain outside the shaded region. In the case of $x_2$, this bound is also tight as the tangent tracks $f$ closely everywhere in the olive region. However, in the case of $x_1$, the bound is loose, as the red line completely decoupled from $f$ throughout a large part of the red region.

The inverse value, $\frac{f'(x)\text{diameter}(\mathbb{D})}{\text{diameter}(\mathbb{F})}$, can be viewed as the number of shaded regions required to cover the entire domain. The NLC is simply the generalization of this concept to multiple dimensions, where the diameter is proxied with the average distance of two points, and the expectation is taken over the data distribution. It is worth noting that the NLC attempts to measure the ratio of diameter of domain and linearly approximable region, *not* the ratio of volumes. Informally speaking, the number of linearly approximable regions required to cover the domain behaves as $NLC^{d_{\text{in}}}$.

### A.2  2D NEURAL NET EXAMPLE

In this section, we illustrate the function computed by neural networks at varying levels of non-linearity. Specifically, in Table 2, we depict the function computed by plain, fully-connected, He-initialized batchnorm-ReLU networks at seven different depths in their randomly initialized state.

We set $d_{\text{out}} = 3$ and set the width of all other layers to 100. We then generated three 100-dimensional Gaussian random inputs $x^{(1)}$, $x^{(2)}$ and $x^{(3)}$. We associated each point $(a, b, c)$ that lies on the unit sphere in $\mathbb{R}^3$, i.e. that has $a^2 + b^2 + c^2 = 1$, with the value $ax^{(1)} + bx^{(2)} + cx^{(3)}$. We call the sphere of points $(a, b, c)$ associated with these inputs the "input sphere".

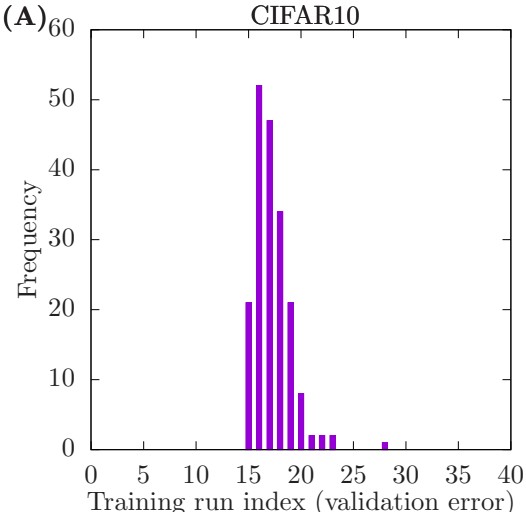
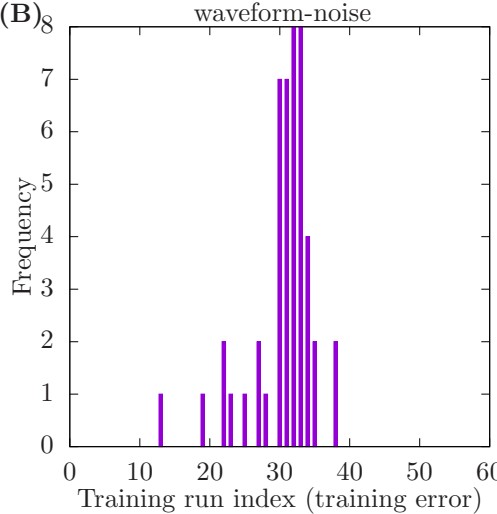

Figure 7: Frequency with which each training run minimized the validation error on CIRAR10 (A) / training error on waveform-noise (B). Note: Architectures which did not achieve a better-than-random test error were omitted in (A) and architectures that did not achieve a better-than-random training error were omitted in (B). We set those thresholds at 80% for CIFAR10 (10 different labels) and 50% for waveform-noise (3 different labels).

We propagate each of those inputs forward through the network. We obtain a 3-dimensional output, which we divide by its length. Now the output lies on the unit sphere in $\mathbb{R}^3$. Each point on that "output sphere" is associated with a color as shown in figure 6. Finally, we color each point on the input sphere according to its respective color on the output sphere.

These colored input spheres are shown in table 2 as azimuthal projections. The RGB values of colors on the output sphere are chosen so that the R component is largest whenever the first output neuron is largest, the G component is largest whenever the second output neuron is largest and the B component is largest whenever the third output neuron is largest. If we imagine that the output is fed into a softmax operation for 3-class classification, then "purer" colors correspond to more confident predictions.

For comparison, we show the NLC on CIFAR10 for batchnorm-ReLU networks of the same depth (median of 10 random initializations). We find that as depth and the NLC of the network increases, the color, and thus the value of the output, change more and more quickly. Of course, this chaotic behavior of the output correspondingly implies smaller linearly approximable regions.

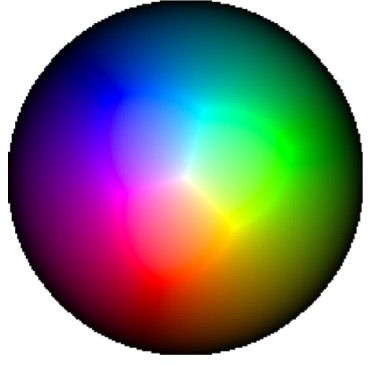

Figure 6: Coloring of the output sphere used for the illustrations in table 2, shown as an azimuthal projection.

# B  LARGE-SCALE EMPIRICAL STUDY - ADDITIONAL RESULTS

In this section, we expand upon findings from our large-scale empirical study that were outlined in section 4.

## B.1  WHAT IS THE IDEAL LEARNING RATE?

One of the hallmarks of our study was the fact that we conducted an exhaustive search over the starting learning rate for training with SGD. We trained our 750 architectures with 40 different

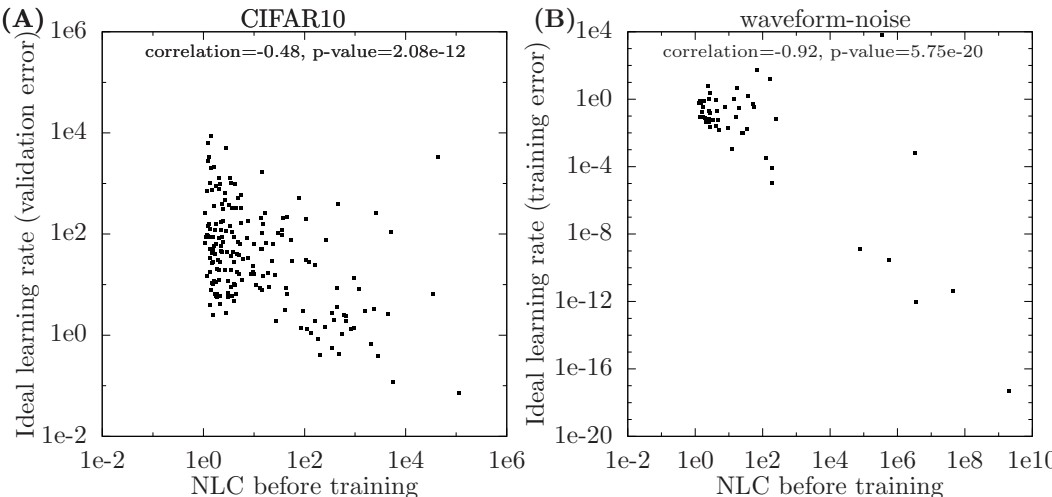

Figure 8: Starting learning rate of the selected training run for minimizing validation error on CI-FAR10 (A) and minimizing training error on waveform-noise (B). Note: Architectures which did not achieve a better-than-random test error were omitted in (A) and architectures that did not achieve a better-than-random training error were omitted in (B). We set those thresholds at 80% for CIFAR10 (10 different labels) and 50% for waveform-noise (3 different labels).

starting learning rates each. Those learning rates formed a geometric sequence with spacing factor 3. The sequence was not the same for each architecture. In fact, the smallest of the 40 learning rates was chosen so that the weight update could still be meaningfully applied in 32 bit precision. See section E.2 for details. Of course, this was simply a heuristic, with the aim of providing a range of learning rates that would contain the ideal learning rate with very high probability.

To verify that this goal was achieved, in figure 7A, we plot a histogram of the index of the training run that yielded the lowest validation error for CIFAR10. The training run with index 1 used the lowest starting learning rate, whereas the training run with index 40 used the largest starting learning rate. Note that we did not plot architectures that did not attain a test error of under 80%, i.e. a non-random test error, as for those architectures the learning rate was not chosen meaningfully. We find that while a wide range of training run indeces were chosen, there was a wide margin on each side of training runs that were never chosen. This is precisely what that confirms that, with high probability, we found the ideal learning rate for each architecture that has the potential to generalize.

We also retrained 50 randomly chosen waveform-noise architectures without applying early stopping based on the validation error. Instead, we continued training to determine the lowest training classification error that could be achieved. The results were plotted in figure 3D. For this experiment, we used 60 training runs. Here, the smallest starting learning rate was chosen so that the weight updates could still be meaningfully applied in 64 bit precision. In figure 7B, we find that indeed the range of training run indeces used is much wider. For 2 architectures, the chosen training run falls outside the range of the original 40 training runs.

We hypothesized that architectures that have very high NLCs and cannot generalize are nonetheless trainable with very small learning rates in 64 bit precision. This is precisely what we find in practice. In figure 8, we plot the NLC in the randomly initialized state against the starting learning rate corresponding to the chosen training run. Figure 8A depicts learning rates which minimized validation error on CIFAR10 and figure 8B depicts learning rates which minimized training error on waveform-noise. In other words, we show the same training runs as in figure 7, and again we removed architectures for which generalization / training failed completely, respectively. While the range of learning rates that lead to good generalization falls in a comparatively smaller range, some architectures can be trained successfully with a learning rate as small as 5e-18!

In general, the reason for this trend is that a large NLC is associated with large gradients, and these gradients need to be down-scaled to keep weight updates bounded. Intriguingly, figure 8B suggests that as the NLC grows, the learning rate should decay as the *square* of the NLC. This observation

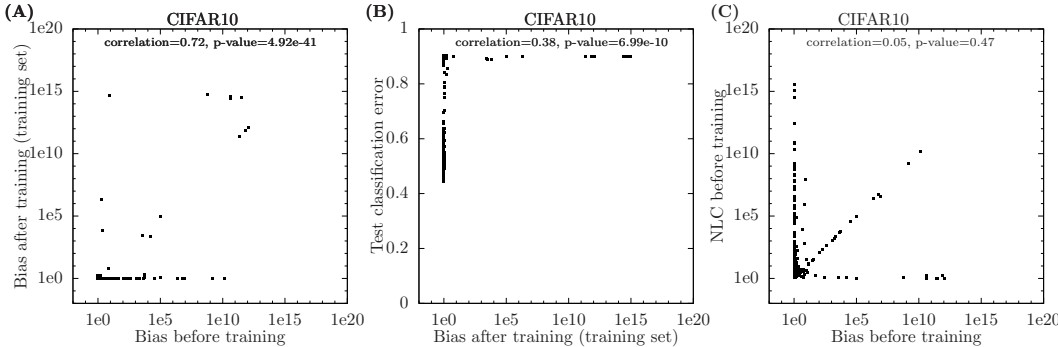

Figure 9: Detailed results from our empirical study. See main text for explanation and section E.2 and E.3 for further details.

mirrors that of Philipp et al. (2018), who observed that the magnitude of weight updates should scale inversely as the gradient increases, which would require the learning rate to scale with the inverse square.

## B.2 THE IMPORTANCE OF AVOIDING EXCESSIVE OUTPUT BIAS

In figure 3A, we show that high bias before training, defined as $\frac{\mathbb{Q}_{j,x}f(x,j)}{\mathbb{Q}_j(\mathbb{S}_x f(x,j))}$, leads to high test error. In figure 9, we investigate this quantity further. In figure 9A, we find that just like the NLC, the bias decreases during training in many cases. In fact, it often reaches a near-zero value. In figure 9B, we find that this is in fact necessary for the network to achieve a better-than-random test error at all. This is not entirely surprising for a dataset like CIFAR10, where each label occurs equally frequently. In figure 9C, we show that many architectures (those near the bottom of the chart) attain a high bias but a low NLC. This confirms that a high makes an independent contribution to test error prediction. All bias values were computed on the training set.

Finally, we note that at the time of writing, we are working on an "improved" version of SGD that can successfully train high-bias architectures and enable them to generalize. Discussing this algorithm, as well as the other signals that exist in figure 9 (many architectures cluster around 1D subspaces in all three graphs ...), unfortunately, goes beyond the scope of this paper.

## B.3 THE VALUE OF USING SKIP CONNECTIONS

In figure 2, we show in blue all architectures that have skip connections, whereas we show in black architectures without skip connections. In that figure, we find that architectures with skip connections not only exhibit a lower NLC overall, but also tend to outperform architectures without skip connections that have similar NLCs.

As it can be hard to distinguish colors in a scatter plot, in figure 10, we plot the results for both types of architectures separately. Both the first row of graphs (A/B/C) and the second row of graphs (D/E/F) are identical to figure 2, except the top row shows only architectures without skip connections and the bottom row shows only architectures with skip connections. The difference in behavior is clear.

## C ARCHITECTURE SAMPLING

In this section, we describe the randomly sampled architectures that we used for our large-scale study.

Each network layer is composed out of a fully-connected linear operation with bias and an activation function. Some architectures have a normalization operation between the linear operation and the activation function. The last layer does not contain an activation function. Some architectures have skip connections, which always bypass two layers as in He et al. (2016). They start after either the

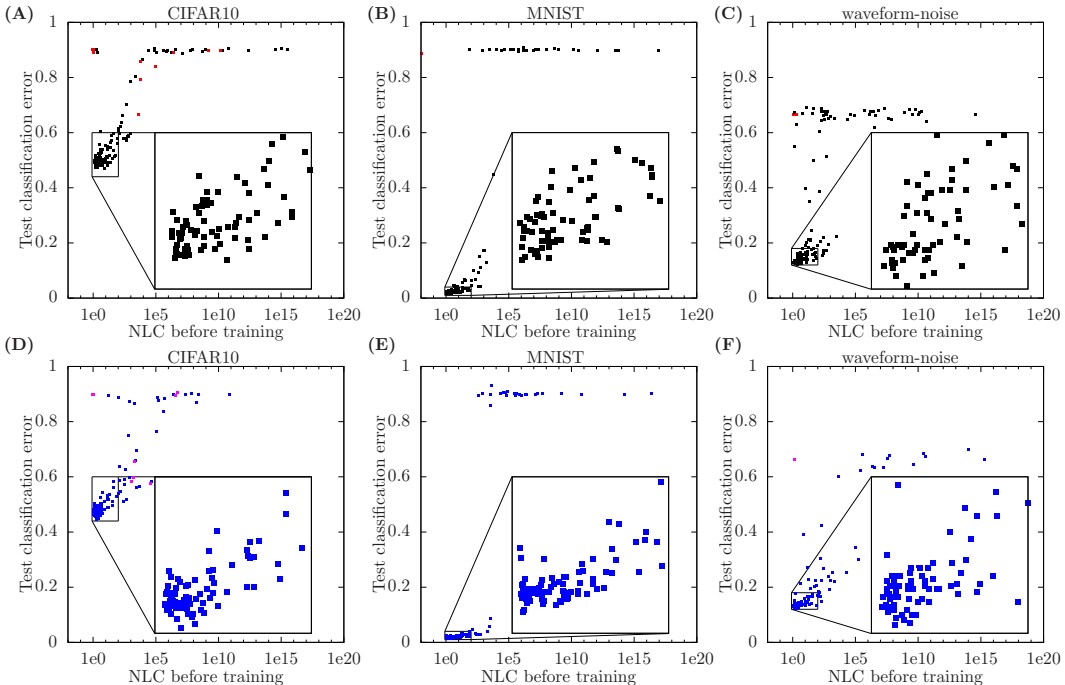

Figure 10: Both the first row of graphs (A/B/C) and the second row of graphs (D/E/F) are identical to figure 2, except the top row shows only architectures without skip connections and the bottom row shows only architectures with skip connections. Again, red color indicates architectures with high output bias.

linear operation or after the normalization operation. They end after the linear operation. The first skip connection begins after the linear or normalization operation in the first layer. The last skip connections ends after the linear operation in the last layer. All skip connections are identity skip connections, except the last skip connection, which has different input and output widths $d_{\text{skip\_in}}$ and $d_{\text{skip\_out}}$ respectively. The last skip connection multiplies the incoming signal with a $d_{\text{skip\_out}} \times d_{\text{skip\_in}}$ submatrix of a $\max(d_{\text{skip\_in}}, d_{\text{skip\_out}}) \times \max(d_{\text{skip\_in}}, d_{\text{skip\_out}})$ uniformly random orthogonal matrix, multiplied by $\max(1, \sqrt{\frac{d_{\text{skip\_out}}}{d_{\text{skip\_in}}}})$. The multiplier is chosen to approximately preserve the scale of the incoming signal in the forward pass. This projection matrix is not trained and remains fixed throughout training.

Each architecture was selected independently at random via the following procedure.

- depth: Depth is chosen uniformly from the set of odd numbers between and including 3 and 49. We used odd numbers to avoid conflicts with our skip connections, each of which bypass two linear operations but do not bypass the first linear operation.

- width: Width was chosen automatically as a function of depth so that the number of trainable parameters in the network is approximately 1 million. The width of all layers except the input and output layer, which are determined by the data, is identical.

- linear operation: A $d_{\text{outgoing}} \times d_{\text{incoming}}$-dimensional weight matrix is initialized as a $d_{\text{outgoing}} \times d_{\text{incoming}}$-submatrix of a $\max(d_{\text{incoming}}, d_{\text{outgoing}}) \times \max(d_{\text{incoming}}, d_{\text{outgoing}})$ uniformly random orthogonal matrix, multiplied by $\max(1, \sqrt{\frac{d_{\text{outgoing}}}{d_{\text{incoming}}}})$. The advantages of orthogonal over Gaussian matrices have been documented by e.g. Saxe et al. (2014); Pennington & Worah (2017); Helfrich et al. (2018); Arjovsky et al. (2016); Xiao et al. (2018); Pennington et al. (2017). We used the multiplier of $\max(1, \sqrt{\frac{d_{\text{outgoing}}}{d_{\text{incoming}}}})$ so that the scale of the signal is approximately preserved as it passes forward through the weight matrix, which is a well-accepted practice for avoiding exponential growth or decay in the forward pass

used in e.g. He initialization (He et al., 2015) and SELU initialization (Klambauer et al., 2017). With a probability of 50%, we initialize the all trainable bias vectors as zero vectors and with a probability of 50%, we initialize their components as independent zero mean Gaussians with a variance of 0.05. We took the 0.05 value from Schoenholz et al. (2017). If the biases are initialized as nonzero, we scale the weight matrices with a factor of $\sqrt{0.95}$ to approximately preserve the scale of the output of the entire linear operation. Finally, with a 25% probability, we then additionally multiply all weight matrices and biases jointly by 0.9 and with a 25% probability, we multiply them jointly by 1.1.

- normalization: With a 50% probability, no normalization is used. With a 25% probability, batch normalization (Ioffe & Szegedy, 2015) is used. With a 25% probability, layer normalization (Ba et al., 2016) is used. Normalization operations do not use trainable bias and variance parameters.

- activation function: We select one of the 8 activation functions shown in figure 1. We select ReLU, SeLU and Gaussian with probability $\frac{2}{11}$ each and tanh, even tanh, sigmoid, square and odd square with probability $\frac{1}{11}$ each. We downweighted the probabilities of tanh, even tanh and sigmoid as we considered them similar. The same holds for square and odd square. After choosing the initial activation function, we added additional modifications. If the initial activation function is $\tau(s)$, we replace it by $c\tau(ds + t) + b$. First, $d$ and $t$ are chosen. $d$ is 1 with a 50% probability, 1.2 with a 25% probability and 0.8 with a 25% probability. $t$ is 0 with a 50% probability, 0.2 with a 25% probability and -0.2 with a 25% probability. Then, with a 50% probability, we set $b$ so that if $s$ follows a unit Gaussian distribution, $\tau(s)$ is unbiased, i.e. $\mathbb{E}_{s\sim\mathcal{N}(0,1)}\tau(s) = 0$. Debiasing follows the example of Arpit et al. (2016). Finally, we always set $c$ so that if $s$ is a unit Gaussian, then $\mathbb{Q}_s\tau(s) = 1$. Again, this follows the principle of avoiding exponential growth / decay in the forward pass as mentioned above. $d$, $b$, $c$ and $t$ are fixed throughout training.

- skip connections: With a 50% probability, no skip connections are used. With a 25% probability, skip connections of strength 1 are used, as is usually done in practice. With a 25% chance, we choose a single value uniformly at random between 0 and 1 and set the strength of all skip connections to that value. With a 50% chance, all skip connections start after the linear operation. With a 50% chance, they start after the normalization operation. We introduced these variations to obtain a more diverse range of NLCs amongst networks with skip connections. Note that normalizing the signal between skip connections rather than only within a skip block reduces the gradient damping of the skip connections for reasons related to the $k$-dilution principle (Philipp et al., 2018).

After sampling, we apply one step of post-processing. All networks that have square or odd square activation functions, or skip connections, that also do not have normalization were assigned either batch normalization or layer normalization with 50% probability each. This is, again, to avoid exponential instability in the forward pass. This post-processing lead to the following changes in aggregate frequencies: no normalization - 20.4%, batchnorm - 39.8%, layer norm - 39.8%.

We sampled 250 architectures for each of three datasets. Results pertaining to those architectures are shown in figures 1, 2, 3, 4, 7, 8 and 9.

We used softmax+cross-entropy as the loss function, as is done in the overwhelming number of practical cases. Crucially, after initializing each architecture, we measured the scale $c$ of activations fed into the loss function, i.e. $c = \mathbb{Q}_{x,j}f(x,j)$. We then had the loss function divide the incoming activations by $c$ before applying softmax. This was done so that the loss functions, which yields very different training dynamics when presented with inputs of different sizes, did not confound the outcomes of our study. We believe that the preference of softmax+cross-entropy for outputs of a certain size has confounded the results of studies in the past. $c$ remained fixed throughout training.

When designing our sampling scheme, we attempted to strike a balance between relevance and diversity. On the one hand, we did not want to include architectures that are pathological for known reasons. We initialized all architectures so that the signal could not grow or decay too quickly in the forward pass. Also, we always used orthogonal initialization. The advantages of orthogonal initialization over Gaussian initialization, at least for fully-connected layers has, in our opinion, been demonstrated to the point where we believe this should be the default going forward.

On the other hand, we introduced many variations such as activation function dilation and shift, and skip connection strength that made our architectures more diverse. While those variations are not necessarily common in practice, we made sure that we never deviated from the "default case" by a large amount in any particular area.

# D   DATASETS

## D.1   SELECTION

We wanted to conduct experiments on three different datasets. First, we chose MNIST and CIFAR10 as they are the two most popular datasets for evaluating deep neural networks, and are small enough so that we could conduct a very large number of training runs with the computational resources we had available. The MNIST dataset is composed of 28 by 28 black and white images of hand-written digits associated with a digit label that is between 0 and 9 (citation: MNIST-dataset). The CIFAR10 dataset is composed of 32 by 32 color images of objects from 10 categories associated with a category label (citation: CIFAR10-dataset).

We decided to choose our third dataset from the UCI repository of machine learning datasets. Klambauer et al. (2017) recently validated the SELU nonlinearity, which has since become popular, on a large number of datasets from this repository. We wanted to choose a dataset that Klambauer et al. (2017) also used. To decide upon the specific dataset, we applied the following filters:

- The most frequent class should not be more than 50% more frequent than the average class.
- The dataset should contain between 1.000 and 100.000 datapoints.
- Datapoints should contain at least 10 features.
- The dataset should not be composed of images, as we already study 2 image datasets.
- The dataset should not contain categorical or very sparse features.
- We only considered datasets that we were actually able to find on the repository website.

After applying all these filters, we were left with two datasets: waveform and waveform-noise. They are very similar. We chose the latter because of the greater number of input features. The inputs of the waveform-noise dataset are composed of wave attributes. Each input is associated with one of three category labels based on the wave type (citation: waveform-noise dataset).

## D.2   PROCESSING

For waveform-noise, we normalized the mean and variance of the features. We processed CIFAR10 via the following procedure.

1. We normalize the mean and variance of each datapoint.
2. We normalize the mean of each feature.
3. Via PCA, we determine the number of dimensions that hold 99% of the variance. That number is 810.
4. We map each datapoint to an 810-dimensional vector via multiplication with a $3072 \times 810$ submatrix of a $3072 \times 3072$ uniformly random orthogonal matrix.
5. Finally, we multiply the entire dataset with a single constant so that we obtain $\mathbb{Q}_{x,i}x(i) = 1$.

We used the exact same procedure for MNIST, except that the number of dimensions of the final dataset was 334 instead of 810.

During preliminary experiments, we found that this pre-processing scheme lead to faster training and lower error values than training on the raw data where only the features are normalized. The reason we designed this scheme in the first place was to reduce input dimensionality so that we could avoid an excessive amount of computation being allocated to the first layer, which would strain our computational budget.

The MNIST dataset contains 60.000 training data points and 10.000 test data points. The training data was randomly split into a training set of size 50.000 and validation set of size 10.000. The CIFAR10 dataset contains 50.000 training data points and 10.000 test data points. The training data was randomly split into a training set of size 40.000 and a validation set of size 10.000. The waveform-noise dataset contains 5.000 data points, which were randomly split into a training set of size 3.000, a validation set of size 1.000 and a test set of size 1.000.

As mentioned, for CIFAR10, our input dimensionality was 810. For MNIST, it was 334. For waveform-noise, it was 40. For CIFAR10 and MNIST, the output dimensionality / number of classes was 10. For waveform-noise, it was 3.

## E  EXPERIMENTAL DETAILS

### E.1  LINEARLY APPROXIMABLE REGIONS STUDY (FIGURE 1)

We begin by defining the relative diameter of a linearly approximable region for a given network $f$, input $x$ from distribution $\mathcal{D}$, input direction $u \in \mathbb{R}^{d_{in}}$ and output direction $v \in \mathbb{R}^{d_{out}}$. Starting from $x$, traversing the input space $k$ times for some fractional value $k$ in the direction of $u$ yields $x'' := x + \frac{ku}{||u||_2 2\mathbb{Q}_i(\mathbb{S}_x x(i))}$. As in the definition of the NLC, we use $2\mathbb{Q}_i(\mathbb{S}_x x(i))$ as a proxy for the diameter of the input space. The radius of the linearly approximable region, relative to the size of the input space, is then the largest value of $k$ such that the linear approximation induced by the Jacobian at $x$ is still close to the true value of $f$ at $x''$. Therefore, we take the *diameter* of the linearly approximable region, relative to the size of the input space, as the largest value of $k$ such that the linear approximation induced by the Jacobian at $x$ is still close to the true value of $f$ at $x' = x + \frac{ku}{||u||_2 \mathbb{Q}_i(\mathbb{S}_x x(i))}$. (Note that $x'$ is simply twice as far from $x$ as $x''$.) Specifically, we define "close" as $\frac{1}{2}(f(x') - f(x))^T v \le (\mathcal{J}(x)(x' - x))^T v \le 2(f(x') - f(x))^T v$. In plain words, the change in function value in the direction of $v$ predicted by the local linear approximation must be at least half and at most 2 times the actual change in function value.

We now generalize this quantity to minibatches, in order for it to be meaningfully defined for networks using batchnorm. Again, consider some network $f$. Now also consider a data batch $X \in \mathbb{R}^{d_{in} \times b}$ containing $b$ randomly drawn datapoints from $\mathcal{D}$. Also consider an input direction matrix $U \in \mathbb{R}^{d_{in} \times b}$ and output direction matrix $V \in \mathbb{R}^{d_{out} \times b}$. Now we define the relative diameter of the linearly approximable region as the largest $k$ such that when setting $X' = X + \frac{kU}{||U||_2 \mathbb{Q}_i(\mathbb{S}_{x \text{ a column of } X} x(i))}$, we have $\frac{1}{2}(f(X') - f(X)).V \le (\mathcal{J}(X)(X' - X)).V \le 2(f(X') - f(X)).V$. Here $f$ can be taken to be applied independently to each column of $X$ if it does not use batchnorm and is taken to be applied jointly to all inputs in $X$ if $f$ does contain batchnorm.

The "largest $k$" is computed by starting with $k = 10^{-9}$ and then checking the condition for increasing $k$ until it fails. The values of $k$ we checked formed a geometric series with spacing factor $10^{\frac{1}{10}}$. We could not reliably check smaller values of $k$ due to numerical underflow, which is why architectures with an NLC less than $10^{-9}$ are not shown in figure 1.

For each architecture, we considered a single random initialization. All values were computed in the randomly initialized state. No training was conducted. We use 10 minibatches of size 250 from the respective dataset and draw 10 random Gaussian $U$ and 10 random Gaussian $V$. We obtain one relative region size value for each of $10 * 10 * 10 = 1000$ configurations. Finally, in figure 1, we report the median across those 1000 values for each architecture.

The NLC is computed as described in section G.

### E.2  PREDICTIVENESS STUDY (FIGURES 2, 3, 4, 7, 8, 9 AND 10)

For each architecture, we considered a single random initialization. We trained them with SGD using minibatches of size 250. To ensure that there is no bias with regards to learning rate, we tuned the starting learning rate independently for each architecture by conducting a large number of training runs with various starting learning rates. A training run is conducted as follows. We train with the starting learning rate until the validation classification error (VCE) has not decreased for 10 epochs. Then we rewind the state of the network by 10 epochs (when the lowest VCE was

achieved), divide the learning rate by 3 and continue training until the validation classification error has not improved for 5 epochs. We divide the learning rate by 3 again, rewind and continue training until the validation classification error has not improved for 5 epochs. This process continues until the step size has been divided by 3 ten times. When the VCE has again not improved for 5 epochs, we rewind one last time and terminate the training run.

For each architecture we completed 40 total training runs with 40 different starting learning rates that form a geometric series with spacing factor 3. For each architecture, the smallest starting learning rate considered was computed as follows. We ran the SGD optimizer for 1 epoch with a learning rate of 1 without actually applying the updates computed. For the weight matrix in each layer, we thus obtained one update per minibatch. Let $\delta W_{lb}$ denote the update obtained for layer $l$ and minibatch $b$ and $W_l$ the initial value of the weight matrix in layer $l$. Finally, we used the value $10^{-8} \sum_l \frac{\mathbb{Q}_b ||\delta W_{lb}||_F}{||W_l||_F}$ as our smallest starting learning rate. The rational behind this choice was that no individual weight matrix update obtained with the smallest starting learning rate would perturb any weight matrix during any iteration by more than approximately $10^{-8}$. We chose $10^{-8}$ specifically so that our smallest starting learning rate would be less than the smallest learning rate that can be meaningfully used under 32 bit precision. Nonetheless, we trained all networks using 64 bit precision.

Of course, this choice of smallest starting learning rate is merely a heuristic. We validated this heuristic by checking that no architecture that obtained a non-random test error attained its lowest validation error with either the smallest five or largest five starting learning rates. This condition was fulfilled for all architectures and datasets. Henceforth, we refer to the 'trained network' as the network that was obtained during the training run that yielded the lowest validation classification error and the 'initial network' as the network in the randomly initialized state.

In figure 7, we show which training runs were used and in figure 8, we show which learning rates were used, plotted against the NLC of the initial network. The NLC was computed as in section G.

In figure 2, we plot the test error of the trained network against the NLC of the initial network, again, computed as in section G. We mark in red all points corresponding to architectures for which $1000 \mathbb{Q}_j (\mathbb{S}_x f(x,j)) < \mathbb{Q}_{j,x} f(x,j)$ for the initial network. We mark in blue all points corresponding to architectures that have skip connections. In figure 4, we plot depth versus test error of the trained network.

In figure 3A, we plot the bias value $\frac{\mathbb{Q}_{j,x} f(x,j)}{\mathbb{Q}_j (\mathbb{S}_x f(x,j))}$ of the initial network against the test error of the trained network. In figure 3B, we plot the NLC of the initial network against the NLC of the trained network. If figure 3C, we plot the NLC of the trained network against the test error of the trained network. In both 3B and 3C, the NLC was computed on the training set. However, the value of the NLC computed on the test set was very similar.

We further compare the bias of the initial network against the bias of the trained network, against test error and against the NLC of the initial network in figure 9. The bias and NLC were always computed on the training set. In figure 10, we break down the results of figure 2 into architectures with skip connections and architectures without skip connections.

We then selected 50 random architectures from our 250 waveform-noise architectures. We then trained these architectures again, with two changes to the protocol: We reduced the learning rate by a factor of 3 only once the training classification error had not been reduced for 10 / 5 epochs respectively; and we considered 60 different step sizes which formed a geometric series with spacing factor 3 and start value $10^{-16} \sum_l \frac{\mathbb{Q}_b ||\delta W_{lb}||_F}{||W_l||_F}$. Therefore, we considered even the smallest step size that was meaningful for 64 bit precision training. This change allowed us to successfully train even architectures with very high NLC. See section B.1 for an analysis on this point. The reason we only trained 50 architectures for this scenario is because training can take a very long time without using the validation set for early stopping, leading to considerable computational expense. The results are presented in figure 3D.

Finally, for figure 3F, we re-trained our 250 waveform-noise architectures with Adam instead of SGD. The protocol was the same (40 training runs), except before obtaining our measurements for $\delta W_{lb}$, we first ran Adam for 4 epochs, again without applying updates, in order to warm-start the

running averages. Only then did we run it for another epoch where we actually gathered values for $\delta W_{lb}$. Again, we verified that the first and last 5 training runs were never used.

### E.3 ERROR ROBUSTNESS STUDY (FIGURE 3)

We computed the maximal error-preserving perturbation shown in figure 3E similarly to the linearly approximable region relative diameter in section E.1. The difference is that instead of requiring that the local linear approximation be close to the true function, we required that the test error over the path from $X$ to $X'$ be at most 5% higher than than the test error at $X$. The test error 'over the path' is defined as the fraction of inputs in the batch that were correctly classified everywhere on the line from $X$ to $X'$. Again, we started with $k = 10^{-9}$ and increased it by $10^{\frac{1}{10}}$ at each step, checking whether each input is correctly or incorrectly classified. We chose the 5% threshold so that architectures with a test error of around 90% on CIFAR10 / MNIST would yield finite outcomes. The values shown in figure 3E are the median over $10 * 10 = 100$ values obtained from 10 random minibatches of size 250 and 10 Gaussian random direction matrices $U$. The random direction matrix $V$ used in section E.1 does not come into play here.

### E.4 APPROXIMABILITY STUDY (TABLE 1)

$NLC_\tau(1)$ was computed as defined in section 5. $NLC_\tau(1)^{48}$ is simply the exponentiated value. The linear approximation error is computed as $\left(\frac{\mathbb{Q}_{s \sim \mathcal{N}(0,1)}(\tau(s) - \bar{\tau}(s))}{\mathbb{Q}_{s \sim \mathcal{N}(0,1)} \bar{\tau}(s)}\right)^2$, where $\bar{\tau}$ is the best linear fit to $\tau$ for inputs drawn from $\mathcal{N}(0,1)$, i.e. $\arg \min_{\bar{\tau} \text{ linear}} \mathbb{Q}_{s \sim \mathcal{N}(0,1)}(\tau(s) - \bar{\tau}(s))$.

NLC was computed as in section G. We show the median across 10 random initializations. The values for different initializations show little variation except for 49-layer networks with square or odd square activation functions.

## F PROOF OF PROPOSITION 1 AND DERIVING THE FORM OF THE NLC

Let $A$ be an $m \times n$ matrix and $u$ a random vector of fixed length and uniformly random orientation. Then we have

$$\mathbb{Q}_u \|Au\|_2$$
$$= \mathbb{Q}_u \sqrt{\sum_{i=1}^m (\sum_{j=1}^n A(i,j)u(j))^2}$$
$$= \mathbb{Q}_u \sqrt{\sum_{i=1}^m \sum_{j=1}^n \sum_{k=1}^n A(i,j)u(j)A(i,k)u(k)}$$
$$= \sqrt{\mathbb{E}_u \sum_{i=1}^m \sum_{j=1}^n \sum_{k=1}^n A(i,j)u(j)A(i,k)u(k)}$$
$$= \sqrt{\sum_{i=1}^m \sum_{j=1}^n \sum_{k=1}^n A(i,j)A(i,k)\mathbb{E}_u u(j)u(k)}$$
$$= \sqrt{\sum_{i=1}^m \sum_{j=1}^n \sum_{k=1}^n A(i,j)A(i,k)\frac{\|u\|_2^2}{n}\delta_{j,k}}$$
$$= \frac{\|u\|_2}{\sqrt{n}} \sqrt{\sum_{i=1}^m \sum_{j=1}^n A(i,j)^2}$$
$$= \frac{\|A\|_F}{\sqrt{n}} \|u\|_2$$

as required.

Further, we have

$$
\mathbb{Q}_x \frac{||\mathcal{J}(x)||_F \mathbb{Q}_{x,x'}||x - x'||_2}{\sqrt{d_{in}} \mathbb{Q}_{x,x'}||f(x) - f(x')||_2}
$$

$$
= \frac{\mathbb{Q}_x ||\mathcal{J}(x)||_F \mathbb{Q}_{x,x'}||x - x'||_2}{\sqrt{d_{in}} \mathbb{Q}_{x,x'}||f(x) - f(x')||_2}
$$

$$
= \frac{\mathbb{Q}_x ||\mathcal{J}(x)||_F \sqrt{\mathbb{E}_{x,x'}||x - x'||_2^2}}{\sqrt{d_{in}} \sqrt{\mathbb{E}_{x,x'}||f(x) - f(x')||_2^2}}
$$

$$
= \frac{\mathbb{Q}_x ||\mathcal{J}(x)||_F \sqrt{d_{\text{in}} \mathbb{E}_i \mathbb{E}_{x,x'}(x(i) - x'(i))^2}}{\sqrt{d_{in}} \sqrt{d_{\text{out}} \mathbb{E}_j \mathbb{E}_{x,x'}(f(x,j) - f(x',j))^2}}
$$

$$
= \frac{\mathbb{Q}_x ||\mathcal{J}(x)||_F \sqrt{\mathbb{E}_i [2\mathbb{E}_x x(i)^2 - 2(\mathbb{E}_x x(i))^2]}}{\sqrt{d_{out}} \sqrt{\mathbb{E}_j [2\mathbb{E}_x f(x,j)^2 - 2(\mathbb{E}_x f(x,j))^2]}}
$$

$$
= \frac{\mathbb{Q}_x ||\mathcal{J}(x)||_F \sqrt{\mathbb{E}_i [\mathbb{E}_x x(i)^2 - (\mathbb{E}_x x(i))^2]}}{\sqrt{d_{out}} \sqrt{\mathbb{E}_j [\mathbb{E}_x f(x,j)^2 - (\mathbb{E}_x f(x,j))^2]}}
$$

$$
= \frac{\mathbb{Q}_x ||\mathcal{J}(x)||_F \mathbb{Q}_i \sqrt{\mathbb{E}_x x(i)^2 - (\mathbb{E}_x x(i))^2}}{\sqrt{d_{out}} \mathbb{Q}_j \sqrt{\mathbb{E}_x f(x,j)^2 - (\mathbb{E}_x f(x,j))^2}}
$$

$$
= \frac{\mathbb{Q}_x ||\mathcal{J}(x)||_F \mathbb{Q}_i (\mathbb{S}_x x(i))}{\sqrt{d_{out}} \mathbb{Q}_j (\mathbb{S}_x f(x,j))}
$$

$$
= NLC(f, \mathcal{D})
$$

Here, both $x$ and $x'$ are drawn independently from $\mathcal{D}$.

## G  COMPUTING THE NLC

Both $\mathbb{Q}_i(\mathbb{S}_x x(i))$ and $\mathbb{Q}_j(\mathbb{S}_x f(x,j))$ can be computed exactly over the dataset in trivial fashion if $f$ does not use batchnorm. If $f$ does use batchnorm, however, $\mathbb{Q}_j(\mathbb{S}_x f(x,j))$ depends on the batch selection. In this case, we replace $\mathbb{Q}_j(\mathbb{S}_x f(x,j))$ in the definition with the NLC by $\mathbb{Q}_j(\mathbb{S}_{X,\beta} f(X,j,\beta))$. Here, $X$ is a data batch matrix of dimensionality $d_{\text{in}} \times b$, where $b$ is the minibatch size, each column of $X$ is an independently drawn input from $\mathcal{D}$ and $\beta$ is uniformly drawn from $\{1,..,b\}$. $f(X,j,\beta)$ is the $(j,\beta)$'th entry of the output of $f$ when $X$ is jointly propagated through the network. In plain terms, we generalize the NLC to batchnorm networks by joining each input $x$ with every possible minibatch $X$. We compute $\mathbb{Q}_j(\mathbb{S}_{X,\beta} f(X,j,\beta))$ in practice by simply dividing the dataset once into minibatches of size 250 and then taking the standard deviation of all output activation values observed during this single pass.

Now we turn our attention to $\mathbb{Q}_x ||\mathcal{J}(x)||_F$. Before we tackle this quantity, we show a property of the Frobenius norm similar to that shown in proposition 1. Let $A$ be a $m \times n$ matrix and let $u$ be an $m$-dimensional unit Gaussian vector. Then we have

$$\mathbb{Q}_u||uA||_2$$

$$= \mathbb{Q}_u\sqrt{\sum_{j=1}^{n}(\sum_{i=1}^{m}u(i)A(i,j))^2}$$

$$= \mathbb{Q}_u\sqrt{\sum_{j=1}^{n}\sum_{i=1}^{m}\sum_{k=1}^{m}u(i)A(i,j)u(k)A(k,j)}$$

$$= \sqrt{\mathbb{E}_u\sum_{j=1}^{n}\sum_{i=1}^{m}\sum_{k=1}^{m}u(i)A(i,j)u(k)A(k,j)}$$

$$= \sqrt{\sum_{j=1}^{n}\sum_{i=1}^{m}\sum_{k=1}^{m}A(i,j)A(k,j)\mathbb{E}_u u(i)u(k)}$$

$$= \sqrt{\sum_{j=1}^{n}\sum_{i=1}^{m}\sum_{k=1}^{m}A(i,j)A(k,j)\delta_{i,k}}$$

$$= \sqrt{\sum_{j=1}^{n}\sum_{i=1}^{m}A(i,j)^2}$$

$$= ||A||_F$$

So, specifically, we have $\mathbb{Q}_x||\mathcal{J}(x)||_F = \mathbb{Q}_{x,u}||u\mathcal{J}(x)||_2$ for unit Gaussian $u$. Therefore, we can estimate $\mathbb{Q}_x||\mathcal{J}(x)||_F$ stochastically by replacing the loss gradient at the output layer with Gaussian random vectors during backpropagation and taking the quadratic expectation over the input gradient. In practice, we compute $\mathbb{Q}_x||\mathcal{J}(x)||_F$ by sampling 100 minibatches of size 250 and clamping independently drawn unit Gaussian random vectors at the output each time.

Finally, let's look at $\mathbb{Q}_x||\mathcal{J}(x)||_F$ for batchnorm networks. Again, this quantity is dependent on batch selection. In fact, in the definition of the NLC, we replace $\mathbb{Q}_x||\mathcal{J}(x)||_F$ by $\frac{1}{\sqrt{b}}\mathbb{Q}_X||\mathcal{J}(X)||_F$, where $\mathcal{J}(X)$ is a $bd_\text{out} \times bd_\text{in}$ matrix that contains the gradient of every component of $f(X)$ with respect to every component of $X$. It is easy to check that this value equals $\mathbb{Q}_x||\mathcal{J}(x)||_F$ for networks without batchnorm. In that case, $\mathcal{J}(X)$ simply takes the form of a block-diagonal matrix with the individual $\mathcal{J}(x)$ from the batch forming the blocks. As before, we have $\mathbb{Q}_X||\mathcal{J}(X)||_F = \mathbb{Q}_{u,X}||u\mathcal{J}(X)||_2$, where $u$ is a $bd_\text{out}$-dimensional unit Gaussian random vector. Hence, just as before, we can stochastically compute $\mathbb{Q}_X||\mathcal{J}(X)||_F$ by sampling random minibatches and backpropagating what is now effectively a $d_\text{out} \times b$-dimensional unit Gaussian matrix, and computing the quadratic expectation on the resulting input gradient. As before, we sample 100 random minibatches of size 250.

