# OpenReview forum: "The Nonlinearity Coefficient - Predicting Generalization in Deep Neural Networks"
_ICLR.cc/2019/Conference_

### Official Review · AnonReviewer2 · 2018-11-01
**Does the nonlinearity coefficient measure nonlinearity?**

**Rating:** 4
**Confidence:** 3

**Review:**

I do not understand the denomination of nonlinearity coefficient provided in definition 1: although the quantity indeed does equal to 1 under whitened data distribution or orthogonal matrix, the conjecture that it should be close to 1 does not seem to be close at all just under any data distribution. Using a similar construction that section 6, we can rescale a whitened input data with a diagonal matrix D with components all equal to one except for a very large one \lambda and also multiply the input weights by D^{-1} to compensate (and have a similar function). If you look at such construction for the linear case with identity initialization of A, the NLC is sqrt((\lambda^2 + n - 1) (\lambda^{-2} + n - 1)) / n which can grow arbitrarily large with \lambda *for a linear model*. However, because of its low capacity, we would expect a linear model to have reasonable generalization. This seems to compromise the initial NLC being low as a necessary condition for reasonable generalization.
Conversely, it’s possible to initialize arbitrarily large residual networks such that the resulting initial function is linear (by initializing the output weight of the incrementing block to 0). This initialization may also be done such that the initial NLC becomes close to 1. I would not think this wouldn’t necessarily result in good generalization, which seems to agree with the experimental observation.
Now given that this initial NLC is neither sufficient nor necessary to predict generalization, one can wonder what is correlating generalization and NLC together in the experiment section. Same remark applies to the correlation between nonlinearity and NLC. This is especially concerning since in the linear case, the NLC can vary whether we chose to whiten the data or not for example, so the other influencing factors need to be discovered. What were the architecture that resulted in small/high NLC?
The experiment section still contains interesting bits, such as successful training of very deep architecture that are very sensitive to input perturbations but they are not part of the main thread of the paper.

---

> ### Author Response · Authors · 2018-11-07
> **Review response (part 2/2)**
>
> "one can wonder what is correlating generalization and NLC together in the experiment section. Same remark applies to the correlation between nonlinearity and NLC."
>
> NLC is a measure of nonlinearity that is based on certain assumptions as explained in sections 3 / A. Figure 1 shows that those assumptions hold in practical networks. Figure 2 shows that good / bad performance is strongly associated with nonlinearity. At least one reason for this relationship is given in figure 3E, where we show that NLC is related to sensitivity of the output to small input changes.
>
> "What were the architecture that resulted in small/high NLC?"
>
> The magnitude of the NLC is chiefly dependent on the linear approximability of nonlinearities, as explained in section 5. Unfortunately, an in-depth discussion on why certain architectures have a certain NLC, for a large number of architectures, goes beyond the scope of the paper.
>
>
> Thank you and we look forward to your response.

---

> ### Author Response · Authors · 2018-11-07
> **Review response (part 1/2)**
>
> Dear reviewer 2,
>
> Thank you for your review.
>
> We believe the standard you measure our paper by is completely unrealistic. You seem to imply that the NLC can only be useful if it is perfect, i.e. if there is no network with an initially high NLC that performs well and no network with an initially low NLC that performs badly. The space of neural networks effectively contains every function that has parameters with respect to which meaningful gradients can be computed. There is no way that a single metric could possibly explain all performance variations across a space that large. The goal of the paper is to show that the NLC is highly predictive of performance in networks built from common design paradigms. But of course the NLC can only account for some of the variance of test error. There are of course factors that influence performance that are not related to nonlinearity. For example, a linear network with a bottleneck layer of width 1 will have a high test error despite being linear. Conversely, any network that has a low training error will also have a low test error if, say, the training and test set happen to be the same, no matter how nonlinear the network is. On an image dataset a ConvNet will achieve lower error than a fully-connected net even if the nonlinearity is the same. Again, a single scalar metric cannot hope to explain all of the performance variation. However, we believe the fact that the NLC can explain a large fraction of that variation across a wide range of networks is very significant.
>
> We strongly believe that the NLC's robustness to confounders and architectural design choices is a *strength* of our paper, not a *weakness*. We improve in these categories upon all the works we cited in our paper, the vast majority of which were published at top conferences (e.g. Balduzzi et al, Cisse et al, Glorot et al, He et al, Raghu et al, Schoenholz et al, Xiao et al, Yang & Schoenholz) Not only do we validate the NLC across a range of networks whose breadth exceeds most if not all related work, it is robust to many failure cases that other metrics are not robust to (section 6). In fact, a core motivation of writing our paper was precisely to advance the state-of-the art in robustness and breadth of applicability. If you care strongly about these things, shouldn't you welcome our paper as improving the state-of-the-art rather than condemning it for not reaching "perfection"? Finally, while our derivation of the NLC in section 3 / A uses heuristics, it is still more principled than the majority of competing metrics.
>
> Regarding the two failure cases you mentioned:
>
> We disagree with your assertion regarding ResNets. Our experience suggests ResNets with zero weights in the last tensor of a residual block generalize well no matter how deep they are. This is supported by "Lechao Xiao, Yasaman Bahri, Jascha Sohl-Dickstein, Samuel S. Schoenholz, and Jeffrey Pennington. Dynamical isometry and a mean field theory of cnns: How to train 10,000-layer vanilla convolutional neural networks.". This paper shows how to train arbitrarily deep networks. While this paper deals with plain nets rather than ResNets, the same principles apply in both cases. Note that as depth increases, we must decrease the learning rate and increase the numerical precision of the computation, as we outline in section B.1 of our paper.
>
> The linear network where the weights with respect to a single input dimension have a much different size than all other weights seems to be a purely theoretical construction. I have never seen such a weight initialization used. Furthermore, this problem is easily fixable. The NLC is currently defined as / equivalent to $\sqrt{\frac{\Tr(\mathbb{E}_x\mathcal{J}\mathcal{J}^T)\Tr(C_x)}{d_\text{in}\Tr(C_f)}}$. Instead, we can define the NLC as $\sqrt{\frac{\Tr(\mathbb{E}_x\mathcal{J}C_x\mathcal{J}^T)}{\Tr(C_f)}}$. Then, the NLC is exactly equal to 1 for all linear functions. The reason we did not use this more complicated definition of the NLC in our paper was because in practice, the weight matrix is initialized without regard to the spectrum of the input. We did not want to make the NLC more complicated without gaining practical benefits. After all, the NLC is supposed to measure nonlinearity for (especially randomly initialized) neural networks, not every possible function.

---

### Official Review · AnonReviewer1 · 2018-11-04
**Solid contribution**

**Rating:** 7
**Confidence:** 4

**Review:**

This paper proposes a metric to measure the "nonlinearity" of neural network, and presents evidence that the value of this metric at initialization time is predictive of generalization performance.

Apart from a few problems I think this paper is well written and thorough. The contribution is solid, although not earth shattering given previous work on such metrics.  There seems to be a basic error in some of the early math, although I don't think this will qualitatively affect the results in any significant way.


-----------------
Detailed comments by section:
------------------

Section 3:

It seems like a 1/sqrt(d) factor is missing from these Q_i(S_x x(i)) and Q_j(S_x f(x,j)) formulas.  As far as I can tell this doesn't affect Def 1 because you seemed to use the correct formula there.

However, the rewritten version with the traces doesn't seem to be correct. There should be a d_in factor in the denominator (inside the square root). This error seems unrelated to the other one.  Assuming I'm correct and that this is an error, does this affect your results in the various figures?  And what is the actual final definition of NLC that you used?

In general, it's annoying for the reader to verify that all of these forms are equivalent.  And it's fiddly enough with the sqrt(d) terms constantly disappearing and reappearing in the numerator and denominators that even you made multiple errors (as far as I can tell).  I would suggest making this section more rigorous and writing out everything carefully. And you probably don't need to rewrite it in so many equivalent forms with different notation unless they are useful somehow.

The use of the Q and S symbols feels superfluous and counterproductive. Standard notation with expectations and squares wouldn't take much more space and would be a lot clearer.


Section 4:

"we plot the relative diameter of the linearly approximable regions of the network as defined in section 3": but you don't seem to define "relative diameter" there. As far as I can tell it's only defined in Appendix E, and this is only mentioned in the caption of figure 1.  It's impossible to interpret this result without knowing precisely what "relative diameter" is.  If you can't afford to describe this in the main paper you should at least mention that it's a different (more expensive) way of estimating the same thing that the NLC estimates.

In Figure 2, are the higher test errors due to the optimizer failing to lower the training error, or due to a greater generalization gap?  I guess the Figure 3 results suggest the latter possibility, which is surprising to me.


What does it mean to have a "very biased output".  What does that inequality mean intuitively?  Should there be an absolute value on the RHS?  It would be much easier to parse it if it were written in plain notation without these S and Q symbols.


Section 6:

"metric also an" -> "metric also has an"

Can you generate a failure case for "correlation information" that doesn't involve Batch Norm layers?  I don't think the authors of those works meant for their results to deal with that.

Note that there are actually a lot of papers going back to the 90s that discussed and proved representational benefits of depth in neural networks.

---

> ### Author Response · Authors · 2018-11-07
> **Review response**
>
> Dear Reviewer 1,
>
> Thank you for your review. We address your detailed comments below. It seems that your main criticism which prevented our paper from attaining a higher rating was your assessment that "The contribution is solid, although not earth shattering given previous work on such metrics." I would love to know more detail regarding this statement. We believe that many properties demonstrated for the NLC throughout the paper (e.g. predictiveness of error when computed before training across a wide range of architectures, predictiveness of nonlinearity, robustness to confounders, relationship to linear approximability of activation functions) are either novel compared to other metrics or at least have not been demonstrated for them. If you are aware of prior work that contradicts those beliefs, we would love to know.
>
>
> .
> .
> .
>
>
> *** missing sqrt(d) factor
>
> You are absolutely correct. Thank you for pointing this out. For what it's worth, we noticed this problem very soon after we submitted the paper and posted a correction on openreview. You can check our comment "typo found", posted on October 3rd, below. We understand that it is annoying to see conflicting definitions, and we apologize for this. The highlighted "Definition 1" at the top of page 3 is correct. We added the alternative definition using traces at the last moment, thereby producing typos. The reason for including the 'correct' d_in / d_out factors in the definition is to avoid susceptibility of the NLC to changes in input dimension / output dimension that do not affect nonlinearity / performance.
>
> In the revision we just uploaded, we have fixed the d_in / d_out typos. If you think we should still also remove the Q/S notation, please let us know and we will upload an additional revision.
>
> *** relative diameter
>
> We added a reference to section E.1 in the main text, where the formal definition of relative diameter is diven. The informal meaning is discussed in section 3.
>
> *** test error vs training error
>
> Yes, the high test error is mostly caused by bad generalization, at least on the waveform-noise dataset. Please see section B.1 for further details on this.
>
> *** Biased output
>
> An individual neuron has biased activation values if the standard deviation of those values is much smaller than the absolute mean. The output bias as defined by the Q over S ratio is a way to average the bias across neurons in the output layer. The quantity can be written as \sqrt{ E_j,x [f(x,j)^2] / (E_j,x [f(x,j)^2] - E_j [[E_x x(i)]^2]) }.
>
> *** Failure cases for correlation information
>
> The problem with correlation information is that it is susceptible to adding and removing constants. Consider a simple example: performing k-means on a dataset is equivalent to performing k-means on that dataset plus a large constant. Adding the constant destroys correlation information, but does not fundamentally alter the quality of the representation, at least as long as the constant is not comparable in size to the largest representable floating point number.
>
> Hence, any bias in the input that is removed by the network confounds correlation information. Batch normalization is just one way to do this. One can also simply initialize the trainable bias in the first layer to eliminate the bias of the dataset.
>
> Similarly, if the network corrupts correlation information by introducing bias, we can for example use an error function (instead of vanila softmax-cross entropy) that compares the output minus the bias to the labels instead of the output itself. Or we could add the same bias to the labels and use an L2 error function, for example.
>
> We agree that the paper that introduced correlation information possibly did not intend to deal with batchnorm. We do not call into question the validity of their results, but simply point out limitations of the metric.
>
> *** Representational benefits of depth
>
> Admittedly, I am not a huge expert on the literature on representational benefits of depth. Since this is not a core topic for the paper, we hope that including 7 citations is sufficient. However, if there are other papers on depth that you think should be cited, please let us know.
>
>
> Thanks

---

> > ### Comment · AnonReviewer1 · 2018-11-16
> > **Re: Review response**
> >
> > Yes my comments about originality were in relation to those other metrics.  Those metrics have properties not demonstrated for yours, and you haven't shown that they don't have the properties that yours apparently does.  For example, your metric involves the expected norm of the Jacobian, which is a quantity that has been studied a lot already.
> >
> > Yes, I do still think you should get rid of the Q/S notation.
> >
> > Regarding "correlation information". In order to have a proper discussion you would need to define precisely what quantity you are talking about. There are various quantities discussed in those papers, some of which don't seem to involve the input distribution at all.
> >
> > Regarding old papers on benefits of depth.  I believe the papers you cite contain some of the references.  For example, the work of Wolfgang Maass and collaborators from the 90s.

---

> > > ### Author Response · Authors · 2018-11-18
> > > **Response**
> > >
> > > Thank you for your comments.
> > >
> > > We agree that some of the other metrics have been demonstrated to be linked to certain quantities. For example, Lipschitz constant has been linked to adversarial robustness and depth to the efficient representation of certain function classes. However, throughout the paper, we  demonstrate a range of favorable properties of the NLC which we think make it stand out among, or at least be complementary to, other metrics.
> > >
> > > We will remove the Q/S notation in the camera-ready version together with other requested changes. (Since the change is purely a formality, I'm assuming this is fine.)
> > >
> > > Regarding correlation information: the underlying claim behind all quantities associated with this concept is that non-preservation of correlations between datapoints leads to high error. In the paper, we argue against this general point rather than any of the individual metrics, for space reasons. While we agree that a deeper discussion would be preferable, there is not enough space for this in the "related work" section.
> > >
> > > We believe we have cited a sufficient number of papers on the topic of depth and are happy to include specific references if requested.

---

### Official Review · AnonReviewer3 · 2018-11-05
**An interesting proposal lacking justification.**

**Rating:** 5
**Confidence:** 4

**Review:**

In this paper the authors introduce a new quantity, the nonlinearity coefficient, and argue that its value at initialization is a useful predictor of test time performance for neural networks. The authors conduct a wide range of experiments over many different network architectures and activation functions to corroborate these results. The authors then extend their method to compute the local nonlinearity of activation functions instead.

I am a bit torn on this paper. I appreciate the direction that the authors have chosen to pursue. The topic of identifying parameters that are predictive of trainability is certainly interesting and has the potential to be quite impactful. Moreover, the breadth of the experiments conducted by the authors is novel and significant. Finally, I find the the overall manner in which the authors have chosen to present their data refreshingly transparent. Together, this leads me to believe that the quantity proposed by the authors might be useful to researchers.

Having said that, I am concerned by the author’s exposition of the nonlinearity coefficient itself. Fundamentally, my concern stems from the fact that it seems a lot of relatively ad-hoc decisions were made in the construction of the nonlinearity coefficient and an insufficiently good job was done to compare it to other measures of nonlinearity.

Specifically, it feels like an extremely weak definition of nonlinearity to say that the linear approximation of a function fails when it produces values that lie outside of the co-domain of the function. Moreover, I feel as though there is already a well defined notion of nonlinearity at a point that could be constructed by reference to the Hessian (or generally by the approximation error induced by truncating the Taylor series after the linear term). I would like to see some comparison between these two methods.

This is made more troubling given that the correlation found by the authors is present but does not seem especially strong. For example, in fig. 2A it seems like the nonlinearity coefficient varies by at least two orders of magnitude in the inset of the figure where the test accuracy really does not seem sensitive to its value. Prior work (for example, [1] from last years ICLR) has shown strong correlations between the Frobenius norm of the Jacobian and test error (see fig. 5 and fig. 6). Since the definition of the nonlinearity coefficient seems somewhat ad-hoc I would love to see a comparison between it and just looking at the Jacobian norm in terms of predicting test accuracy.

[1] - SENSITIVITY AND GENERALIZATION IN NEURAL NETWORKS: AN EMPIRICAL STUDY
Roman Novak, Yasaman Bahri, Daniel A. Abolafia, Jeffrey Pennington, Jascha Sohl-Dickstein

---

> ### Author Response · Authors · 2018-11-07
> **Review response (part 2/2)**
>
> ### The method of Novak et al ###
>
> Our paper goes significantly beyond the scope of Novak et al, because we use the NLC computed *before* training to predict performance *after* training. Novak et al use the Jacobian *after* training to compare against performance *after* training. Predicting performance before training is much more useful because it enables architecture design / selection. Furthermore, it is also much harder. We predict the property (test error) of one network (trained network) by examining a property (NLC) of a different network (untrained network). Novak et al make inference about the property of a network (test error) from properties of that same network.
>
> Let us detail just one reason why our task is harder. The Novak et al paper uses the Frobenius norm of the Jacobian of the softmax units with respect to the input and compares that value to test error. We can write that Jacobian as d softmax/d input, which is the same as d softmax/d logits * d logits/d input, where 'logits' denotes the values that are fed into the softmax. Now it turns out that ||d softmax/d logits||_F tends to be smaller for a given input when the prediction of the network is correct for that input, and it tends to be larger when the prediction of the network is incorrect. This is shown in the Novak paper in figure 6. Therefore ||d softmax/d logits||_F is strongly correlated with error not because of an interesting structural property of a network, but simply because of an idiosyncratic property of the softmax: it tends to have larger gradients for less confident predictions. Hence, it is likely that the correlation between d softmax/d logits = d softmax/d logits * d logits/d input and test error is also caused to a significant degree by this effect. By computing the NLC before training, we do not "benefit" from this spurious signal.
>
> Therefore, our task is not comparable to the task studied by Novak et al, and hence the raw correlation numbers are also not comparable. We would argue that if you consider the Novak paper to be an important contribution, our paper is at least an equally important contribution, because we study a task that is at least in certain ways significantly more useful.
>
> ### Summary ###
>
> The NLC is the first gradient-based metric that, when computed before traning, has been shown to be predictive of test error after training through a large-scale study involving a wide variety of networks. Additional benefits include:
>
> - it is an accurate measure of nonlinearity *in practice* (figure 1)
> - it is intimiately related to the linear approximability of activation functions (section 5)
> - it is more robust to confounders than comparable metrics (section 6)
> - it is cheap to compute (section G)
>
> However, we do not claim that the NLC is the *correct* measure of nonlinearity for deep neural networks. We are happy to use heuristics in deriving the NLC as long as we attain the above benefits. In response to your criticism that "it feels like an extremely weak definition of nonlinearity to say that the linear approximation of a function fails when it produces values that lie outside of the co-domain of the function." we would respond that our goal is not to define nonlinearity definitively, but to come up with a metric that has the benefits outlined above. Nonetheless, we think that our figure 1 and the shortcomings of the Hessian indicate that the NLC is the state-of-the-art in neural network nonlinearity estimation, and we think that the NLC as a performance predictor is better motivated than e.g. 'gradient explosion' or 'correlation preservation', as well as the metric used by Novak et al.
>
> Finally, we strongly disagree with the statement that "in fig. 2A it seems like the nonlinearity coefficient varies by at least two orders of magnitude in the inset of the figure where the test accuracy really does not seem sensitive to its value." See the correlation numbers and p-values below. While the correlation in the inset is a bit lower, this is to be expected as the correlation between random variables tends to decrease if the range of one variable is restricted.
>
> Scenario correlation p-value
> CIFAR10  0.72 2.34e-41
> CIFAR10 (inset) 0.68 1.88e-21
> MNIST 0.81 1.31e-57
> MNIST (inset) 0.63 3.60e-19
> wave 0.67 1.82e-33
> wave (inset) 0.57 3.60e-13
>
> We thank you again for your review. We would love to discuss further and look forward to your response. Please let us know whether you want us to include the above discussions in the next revision of the manuscript or not.

---

> > ### Comment · AnonReviewer3 · 2018-11-09
> > **Reply**
> >
> > Thanks for your quick and detailed reply, however it did not really address the issues that I raised in my review.
> >
> > My purpose in bringing up the Hessian was not to suggest that it would be a useful metric in practice, and my reason for bringing up the work of Novak et al. was not to say that they had similar results. In your work, you propose “The Nonlinearity Coefficient” (a name that you chose) along with arguments that suggest that it captures nonlinearity. My point was that 1) you have not validated that the NLC actually captures nonlinearity to a sufficient degree and 2) you have not validated that the NLC works better than other measures. Together, these issues lead me to be concerned that your results will mislead practitioners and might do more harm than good.
> >
> > Let me address these two points separately.
> >
> > 1) I agree that the Hessian is intractable for most networks used in practice. To a lesser extent I agree that the Hessian is poorly behaved for ReLU networks (in fact, I think if you add small amounts of Gaussian noise to data it is well-defined). However, for small networks where the Hessian is well-defined and tractable it would offer an appealing way to test the NLC, but you have not done that. Moreover, conceptually the Hessian also provides a good way to think curvature that shows the “relative diameter” you introduce to be a poor measure of nonlinearity when comparing with the NLC.
> >
> > Your only evidence that the NLC captures curvature as far as I can tell is fig. 1. Your definition for diameter is the maximum k such that 1/2(f(x’) - f(x))^Tv < (J(x)(x’-x))^Tv < 2(f(x’) - f(x))^Tv where x’ = x + C k u. Let us now consider this condition a bit more carefully. Suppose that f(x) has a well-defined Taylor series then A(f(x’) - f(x)) = A(J(x) (x’-x) + 1/2(x’-x)H(x)(x’-x) + O((x’-x)^3)). If we assume that u and v are randomly oriented wrt the Hessian and the Jacobian then will have equality between A(f(x’) - f(x))^Tv and (J(x)(x’-x))^Tv when k ~ ||J(x)||_F/||H(x)||_F with some additional constants. It follows that your definition for the diameter ought to be proportional to the Jacobian and inversely proportional to the NLC. This is true regardless of whether or not the NLC is related to nonlinearity at all and is only based on the fact that the metric you chose for diameter depends on the norm of the Jacobian.
> >
> > So what would a more appropriate definition have been? If you had instead considered the k where (f(x’) - f(x) - J(x)(x’-x))^2 > C for some C. Expanding to second order you would expect k ~ C^{1/4} / ||H(x)||_F which would not have this implicit factor of the Jacobian. Plotting this against the NLC would, in my eyes, be a much more rigorous way of checking whether the NLC actually captures anything about curvature.
> >
> > 2) Given that you have not validated that the NLC actually contains information about curvature, it is natural to ask whether or not you are gaining anything aside from the Jacobian. While it is true that the results of Novak et al. were for trained networks, it is also true that they saw strong correlation between the norm of the Jacobian and generalization error. To me, since the derivation of the NLC is suspect, it is all the more important that you check how well the norm of the Jacobian correlates with generalization error (at initialization).
> >
> > Actually, to that end I would like to point out something else that I found unfortunate about your exposition. In sec. 6 you compare the NLC with a number of other metrics that one might want to correlate with generalization performance. However, you do not actually do the comparison, but instead give qualitative justification for why each might not work. This would be more impactful if you actually showed that the NLC performs better than these other metrics. To me this is all the more true given that the NLC itself is poorly justified.
> >
> > As a final point, I have to remark on the table that you presented. I don’t know that I find correlation to be a particularly compelling metric for success in this case (and I don’t know exactly how to interpret a correlation of 0.57, for example). I think a better metric would be to treat this as a regression task and compute the RMSE for a linear model based on the NLC compared with the standard deviation of the data (only for data in the inset). In this way we can see how much the NLC buys you in terms of predicting generalization performance over not doing the prediction. Looking at the inset, I would be very dubious as a machine learning practitioner at preemptively throwing out networks in that range of NLC. At the point where I would be unwilling to do that it seems that the NLC could at most be used to cull trivially poor networks without training (although there is some value in this).
> >
> > I do feel that this could be a solid submission down the road, but in my opinion these are serious issues that must be addressed before the results seem trustworthy.

---

> > > ### Comment · AnonReviewer3 · 2018-11-11
> > > **Addendum**
> > >
> > > I would just like to make a slight modification to my reply. While I do believe that RMSE would be more appropriate than correlation for this particular task, I do not think that I was correct in saying that the NLC could be used to cull only trivially poor networks. The only reason why some networks appear trivially poor is because of the NLC in the first place, which is a success!! As I mentioned in my original review I do overall like your raw results. Nonetheless, I do stand by the rest of the comments in my reply. I do hope that you make the exposition stronger and incorporate baselines in future versions of the manuscript.

---

> > > > ### Author Response · Authors · 2018-11-13
> > > > **Response**
> > > >
> > > > Thank you for your reply.
> > > >
> > > > ### 1) Hessian / curvature
> > > >
> > > > "Your only evidence that the NLC captures curvature as far as I can tell is fig. 1."
> > > >
> > > > "Plotting this against the NLC would, in my eyes, be a much more rigorous way of checking whether the NLC actually captures anything about curvature."
> > > >
> > > > But we are not trying to argue that the NLC captures curvature. We *define* nonlinearity as the inverse relative diameter of linearly approximable regions. We think it is very reasonable to define "linearity" as the size of linearly approximable regions, and therefore "nonlinearity" as the inverse of that. As far as we know, "nonlinearity" does not have an accepted rigorous definition for deep networks, therefore we give it one. The definition of the NLC is well-justified as a measure of the relative diameter, both conceptually (section 3 / A) and experimentally (figure 1).
> > > >
> > > > I don't understand why there is a burden to define the word "nonlinearity" as a synonym to curvature, or why we have the burden to discuss curvature in our paper at all. Our paper is not about curvature. If we had called the "nonlinearity coefficient" instead "linear region coefficient", would you still object to the paper?
> > > >
> > > > "It follows that your definition for the diameter ought to be proportional to the Jacobian and inversely proportional to the NLC."
> > > >
> > > > I'm not 100% sure how you get to this statement, but it is clearly incorrect. While the relative diameter is indeed inversely proportional to the NLC, it is *also* inversely proportional to the Jacobian. The NLC, by definition, is proportional to the Jacobian. Therefore the relative diameter has a similar relationship with the Jacobian as it has with the NLC.
> > > >
> > > > The fact that the NLC approximates the relative diameter is a highly non-trivial observation and cannot be shown with a back-of-the envelope calculation. "k ~ ||J(x)||_F/||H(x)||_F, therefore k is proportional to the Jacobian" makes no sense in practice, because the Jacobian and Hessian are highly dependent. As I mentioned in my original rebuttal, curvature and relative diameter are not directly related, so looking at the Hessian does not help in determining the relationship between NLC and relative diameter.
> > > >
> > > >
> > > > ### 2) NLC vs Jacobian
> > > >
> > > > The pure Jacobian measured before training correlates much less with error than the NLC. On CIFAR10, the correlation is only 0.38. On waveform-noise, it is only 0.44. We are happy to include those numbers in the paper. However, such comparisons are ultimatetely uninteresting for 2 reasons: (1) Because the low correlation of the Jacobian is largely caused by a small number of outliers. Because of the susceptibility of the correlation measure, as well as the RMSE measure, to outliers, it does not lend itself for ranking prediction metrics. We include the correlation in the paper simply in order to statistically verify the existence of the strong relationship between NLC and test error that is evident from figure 2. (2) because the reason the Jacobian is correlated with error is because the Jacobian is correlated with the NLC, by definition. Any overlap in correlation is caused by the trivial insight that the NLC takes into account the Jacobian in its computation.
> > > >
> > > > The reason to prefer the NLC over the pure Jacobian is precisely because the NLC is well-justified: see section 3, A, 5, 6 and figure 1. As far as we can tell, Novak et al provided little to no justification for caring about the Frobenius norm of the raw Jacobian apart from its predictiveness. It is this theoretical grounding of the NLC that makes it robust to the confounders discussed in section 6, to which the raw jacobian is not robust. We improve upon Novak with regards to theoretical justification / grounding, robustness, as well as usefulness of task studied.
> > > >
> > > > Finally, aren't the RMSE measure (relative to the standard deviation of the error) and correlation equivalent? The square of the correlation measures exactly the fraction of variance explained by the predictor via a linear model. Isn't this what you are suggesting?
> > > >
> > > >
> > > >
> > > > Thank you again and we look forward to continuing the discussion.

---

> ### Author Response · Authors · 2018-11-07
> **Review response (part 1/2)**
>
> Dear Reviewer 3,
>
> Thanks you for your review. We agree that the Hessian, as well as the method used in Novak et al, are interesting to consider given the context of our paper. In fact, we considered discussing both them in the paper itself, but ultimately decided to focus only on the 5 metrics discussed in section 6, for space reasons. Please see the discussion of the Hessian and Novak paper below.
>
> ### Using the Hessian ###
>
> The Hessian has a severe drawback for estimating nonlinearity: it fails completely for nonlinearities that are non-differentiable. For example, the input-output Hessian of a plain ReLU or hard tanh network is the zero tensor. Even in a batchnorm-ReLU network, the Hessian treats ReLU's as if they are linear functions and thereby grossly underestimates nonlinearity. Since ReLU is by far the most popular activation function, any nonlinearity measure would certainly have to work for ReLU to be practically useful.
>
> One might approximate ReLU with a smooth function, such as 1/k log(1 + exp(kx)) with a large k, but this has further pitfalls. Consider the Hessian of a nonlinearity operation \tau that is applied elementwise to a vector of pre-activations x with respect to that vector: H_\tau = d^2\tau / dx^2. The off-diagonal entries of this 3D tensor are zero. If \tau is a close approximation to ReLU, then the diagonal entries are either very large if the correspondind entry of x is close to zero, or very small if the corresponding entry of x is not close to zero. Therefore the diagonal entries of H_\tau are very sparse, and hence have high variance. Further, as \tau gets closer and closer to ReLU, ||H_\tau||_F converges to infinity. Finally, because of the chain rule for second derivatives, an infinite / noisy H_\tau causes the Frobenius norm of the input-output Hessian of the network to also be infinite / noisy, yielding to an incorrect / high-variance estimate of nonlinearity.
>
> While ReLU is a prominent failure case, these issues extend far wider. Consider an arbitrary activation function \tau. Then we can approximate it by a piecewise linear activation fucntion \tau'. If the length of the linear segments used is sufficently small, we can replace \tau with \tau' in any network and obtain not just the same learning behavior and final performance, but also the same linearly approximable regions. Yet the Hessian of the network using \tau' grossly underestimates nonlinearity for the reasons outlined above. In fact, using similar constructions, we can maintain the nonlinearity and performance of a network but vary the Hessian almost arbitrarily. This begs the question: amongst this large space of possible Hessians, which is the correct one?
>
> In addition to the conceptual issues, the input-output Hessian of a network is a 3D tensor. We are aware of no algorithm that computes its Frobenius norm efficiently.
>
> In summary, we argue that the Hessian is either outright unsuitable for nonlinearity estimation / performance prediction in deep networks, or significantly more work would need to be done to fix the issues above. In either case, the NLC is the state-of-the art for nonlinearity estimation.

---

### Author Response · Authors · 2018-10-04
**Typo found**

On page 3, the terms that involve the trace operator Tr() are missing some d_in/d_out values. root(Tr(C_x)) should instead be root(Tr(C_x)/d_in). root(Tr(C_f)) should instead be root(Tr(C_f)/d_out). Finally, the two terms that have Tr(E_x JJ^T) and Tr(AA^T) in the numerator should also have d_in in the denominator, under the square root. Otherwise, the text remains unchanged. Apologies for any inconvenience caused.

---

### Meta-Review · Area_Chair1 · 2018-12-16
**Interesting direction but needs more work**

**Confidence:** 4
**Recommendation:** Reject

**Metareview:**

This paper proposes the NonLinearity Coefficient (NLC), a metric which aims to predicts test-time performance of neural networks at initialization. The idea is interesting and novel, and has clear practical implications. Reviewers unanimously agreed that the direction is a worthwhile one to pursue. However, several reviewers also raised concerns about how well-justified the method is: in particular, Reviewer 3 believes that a quantitative comparison to the related work is necessary, and takes issue with the motivation for being ad-hoc. Reviewer 2 also is concerned about the soundness of the coefficient in truly measuring nonlinearity.

These concerns make it clear that the paper needs more work before it can be published. And, in particular, addressing the reviewers' concerns and providing proper comparison to related works will go a long way in that direction.